# A specific role for importin-5 and NASP in the import and nuclear hand-off of monomeric H3

Alonso Javier Pardal, Andrew James Bowman*

Division of Biomedical Sciences, Warwick Medical School, University of Warwick, Coventry, United Kingdom

**Abstract** Core histones package chromosomal DNA and regulate genomic transactions, with their nuclear import and deposition involving importin-β proteins and a dedicated repertoire of histone chaperones. Previously, a histone H3-H4 dimer has been isolated bound to importin-4 (Imp4) and the chaperone ASF1, suggesting that H3 and H4 fold together in the cytoplasm before nuclear import. However, other studies have shown the existence of monomeric H3 in the nucleus, indicating a post-import folding pathway. Here, we report that the predominant importin associated with cytoplasmic H3 is importin-5 (Imp5), which hands off its monomeric cargo to nuclear sNASP. Imp5, in contrast to Imp4, binds to both H3 and H4 containing constitutively monomeric mutations and binds to newly synthesised, monomeric H3 tethered in the cytoplasm. Constitutively monomeric H3 retains its interaction with NASP, whereas monomeric H4 retains interactions specifically with HAT1 and RBBP7. High-resolution separation of NASP interactors shows the 's' isoform but not the 't' isoform associates with monomeric H3, whilst both isoforms associate with H3-H4 dimers in at least three discrete multi-chaperoning complexes. In vitro binding experiments show mutual exclusivity between sNASP and Imp5 in binding H3, suggesting direct competition for interaction sites, with the GTP-bound form of Ran required for histone transfer. Finally, using pulse-chase analysis, we show that cytoplasm-tethered histones do not interact with endogenous NASP until they reach the nucleus, whereupon they bind rapidly. We propose an Imp5-specific import pathway for monomeric H3 that hands off to sNASP in the nucleus, with a parallel H4 pathway involving Imp5 and the HAT1-RBBP7 complex, followed by nuclear folding and hand-off to deposition factors.

*For correspondence:
a.bowman.1@warwick.ac.uk

**Competing interest:** The authors declare that no competing interests exist.

## Editor's evaluation

The authors provide good evidence showing that a pool of newly synthesised H3 and H4 histones are maintained in a monomeric conformation in the cytoplasm and are translocated into the nucleus via the importin-5 protein. The analysis makes great use of various technologies, and the results are certainly interesting.

## Introduction

Histones are one of the most abundant class of proteins within the cell, with nuclear import and deposition into chromatin being a major challenge during S-phase (*Hammond et al., 2017*; *Pardal et al., 2019*). Import, folding, and delivery of histones to sites of incorporation in a timely manner can thus impact a broad range of genomic processes and are of great current interest (*Burgess and Zhang, 2013*; *Filipescu et al., 2014*; *Hammond et al., 2017*; *Gurard-Levin and Almouzni, 2014*; *Mendiratta et al., 2019*).

Histones are synthesised in the cytoplasm and translocate to the nucleus associated with importin proteins (also known as karyopherins) (*Bernardes and Chook, 2020*). Once in the nucleus, importins interact with RanGTP, leading to the unloading of their cargo (*Chook and Süel, 2011*). Work on different organisms has cemented the notion that, despite certain redundancy, most importin–histone interactions are specific (*Baake et al., 2001*; *Mühlhäusser et al., 2001*; *Mosammaparast et al., 2001*; *Mosammaparast et al., 2002*; *Blackwell et al., 2007*). For instance, H2A and H2B have been shown to associate with importin-9 (Imp9) (*Padavannil et al., 2019*; *Mosammaparast et al., 2001*), whereas H3 and H4 have been shown to associate with Imp4 (*Campos et al., 2015*; *Tagami et al., 2004*; *Alvarez et al., 2011*) and, to a lesser extent, Imp5 (*Alvarez et al., 2011*). Similar selectivity occurs in yeast, with H3 and H4 shown to interact with homologs Kap123 and Kap121, respectively (*Mosammaparast et al., 2002*). Interestingly, in vitro H3-H4 have broader binding profiles (*Soniat et al., 2016*) and can be imported by a number of importins in reconstituted systems using isolated nuclei (*Mühlhäusser et al., 2001*).

Histone chaperones also associate promptly with newly synthesised histones, providing an environment for solubility and ensuring correct folding before targeting towards sites of DNA deposition (*Hammond et al., 2017*; *Pardal et al., 2019*; *Menditta et al., 2019*). Numerous factors have been isolated in complex with H3 and H4 from cytosolic extracts (*Groth et al., 2011*; *Tagami et al., 2004*; *Campos et al., 2015*; *Parthun et al., 2011*); however, as many of these factors are overwhelmingly found in the nucleus under native conditions, it is not clear how well cytosolic extracts represent the true cytoplasm of the cell (*Apta-Smith et al., 2018*; *Paine et al., 1992*; *Paine et al., 1983*).

Nuclear autoantigenic sperm protein was first identified as an H1-binding protein expressed in testis (tNASP) (*O'Rand et al., 1992*) that also interacted with HSP90 (*Alekseev et al., 2005*). In dividing somatic cells, a shorter splice isoform is prevalent (sNASP) (*Richardson et al., 2000*), whereas both isoforms often occur in transformed cell lines (*Tagami et al., 2004*). Subsequently, NASP has been shown to interact predominantly with H3 and H4 (*Tagami et al., 2004*; *Groth et al., 2011*; *Campos and Reinberg, 2010*; *Cook et al., 2011*). NASP displays multivalent interactions with its histone cargo, binding to a peptide epitope at the C-terminus of H3 via a canonical TPR–peptide interaction (*Bowman et al., 2016*; *Bao et al., 2022*), and to an N-terminal region of H3 via a site on the surface of the TPR domain (*Cook et al., 2011*; *Bao et al., 2022*; *Liu et al., 2021*). The latter interaction is structurally compatible with H3-H4 dimer binding in conjunction with ASF1. However, the TPR-H3 C-terminal peptide interaction is structurally exclusive (*Bowman et al., 2017*; *Bao et al., 2022*; *Liu et al., 2021*). These structural analyses may partly explain why NASP binds a superstoichiometric amount of H3 over H4; that is, NASP binds to both an H3 monomeric and an H3-H4 dimeric pool of histones (*Apta-Smith et al., 2018*).

Despite significant investigation, it is still not clear where, and in association with what factors, folding of newly synthesised H3-H4 occurs in the cell (*Pardal et al., 2019*). Isolation of an H3-H4 dimer bound to Imp4 and ASF1 in cytosolic extracts has been accepted as evidence for H3 and H4 folding in the cytoplasm (*Tagami et al., 2004*; *Alvarez et al., 2011*; *Campos et al., 2015*). However, it has been shown that such extracts often contain soluble nucleoplasmic components due to the difficulty in rupturing the plasma membrane whilst leaving the nuclear membrane intact (*Apta-Smith et al., 2018*; *Paine et al., 1992*; *Paine et al., 1983*), and thus how these extracts relate to the cytoplasm is not clear. H3 and H4 tethered in the cytoplasm immediately after translation remain monomeric and do not fold with their endogenous counterpart (*Apta-Smith et al., 2018*), whilst superstoichiometric binding of H3 by NASP suggests the presence of a stable monomeric pool of H3 (*Apta-Smith et al., 2018*). Furthermore, monomeric histones serve as effective substrates for nuclear import in isolated nuclei (*Mühlhäusser et al., 2001*), suggesting a monomeric substrate can be recognised by importins, at least in permeabilised cells.

In this study, we investigate the interactome of histones H3 and H4 using constitutive monomeric mutants and post-synthesis trapping using the RAPID-release approach. We identify Imp5 as a primary binder of monomeric H3 and H4 in the cytoplasm, with histone-specific interactions maintained by a number of histone chaperones. Fractionation of NASP-associated complexes revealed a stable NASP-H3 monomer population, roughly equal in size to that of NASP-H3-H4-containing complexes. NASP and Imp5 binding are mutually exclusive, with in vitro experiments showing RanGTP is necessary for hand-off to occur. Lastly, we show that NASP associates with H3 rapidly after its import, suggesting it is a receptor for monomeric H3 after transfer from Imp5. These findings support a model in which

newly synthesised H3 and H4 are imported as monomers bound to Imp5 and fold in the nucleus once associated with their histone chaperones.

## Results

### Constitutively monomeric H3 and H4 translocate to the nucleus and retain binding to a subset of histone chaperones and importins

To probe factors that interact with monomeric histones, we designed mutations that preclude folding of the heterodimer, trapping histones in their otherwise transient, post-synthesis state. As any mutation may omit potential binders, two divergent strategies were taken: (1) breaking the α2 helix of the histone fold domain by insertion of three glycine residues (helix-breaker mutations [HB]) and (2)- disruption of the histone fold by substituting three hydrophobic core residues (fold-disruptor mutations [FD]) (*Figure 1A and B*).

To test their cellular localisation, histone mutants were fused to eGFP and transiently expressed in HEK293-F cells (*Figure 1C*). Both sets of mutants showed nuclear fluorescence in interphase cells, demonstrating that they are import-competent, but did not colocalise to mitotic chromosomes, suggesting that they are not incorporated into chromatin (*Figure 1C*). We further tested chromatin incorporation using fluorescence recovery after photo-bleaching (FRAP) analysis (*Figure 1D*). Chromatinised histones should turnover very slowly, whilst soluble histones should recover almost immediately. Indeed, we found that while recovery of WT histones in the bleached region was slow, recovery of mutant histones occurred within seconds. We therefore conclude that non-dimerising mutants of H3 and H4 are imported into the nucleus, but are not incorporated into chromatin.

To test whether the mutants can fold with their endogenous counterpart, we purified C-terminal eGFP fusions using single-step, GFP-Trap pulldowns (*Figure 1E*), using eGFP fluorescence to ensure equal loading (*Figure 1—figure supplement 1*). As endogenous histones are unusually small, they are well-resolved from other interacting proteins and can be unambiguously identified by Coomassie staining after SDS-PAGE. Whilst WT H3.1-eGFP pulled down H4, and H4-eGFP pulled down H3, mutant histones failed to pull down any of their endogenous counterparts (*Figure 1E*).

To further investigate the associations of mutant histones, we performed Western blots, probing the pulldowns with antibodies for a number of known interacting factors (*Figure 1F*). Whilst WT H3 interacted with the full cohort of probed histone chaperones, H3 mutants only interacted with NASP (both s and t isoforms) and Imp5, but surprisingly not Imp4 (*Figure 1F*). H4 similarly lost the majority of its interacting proteins, retaining only the HAT1 complex; HAT1 and RBBP7 (RbAp46) and Imp5, but not Imp4 (*Figure 1F*). Taken together, these findings show that constitutively monomeric mutants are imported into the nucleus, show differences in their chaperone profiles, but share the common binding of Imp5 over Imp4.

### Interactomes of histone monomer mutants

To unbiasedly identify factors interacting with monomeric H3 and H4, we performed affinity purification followed by mass spectrometry (*Figure 2A–D*, *Figure 2—source data 1 and 2*). Plotting normalised total precursor intensity of WT against mutant, we confirmed Imp5 and NASP retains interaction with monomeric H3, and Imp5 and HAT1-RBBP7 retains interaction with monomeric H4 (*Figure 2A–D*). Notably, binding profiles are similar across both types of mutant (HD or FB) and reproducible across three biological replicates (*Figure 2E and F*).

In addition, we identified a number of HSP70 family members associating with both monomeric and WT H3 (HSPA1B, HSPA6, HSPA8) and H4 (HSPA1B, HSPA8), confirming previously identified interactions (*Campos et al., 2015*) and highlighting the cross-talk between heat shock chaperones and the histone chaperoning network (*Hammond et al., 2021*). The HSP70-interacting protein DNAJC9 showed a strong preference for WT histones in both H3 and H4 pulldowns, as would be expected from its recently solved structure with an H3-H4 dimer (*Hammond et al., 2021*). Similarly, factors involved in the direct deposition of histones onto DNA (CAF1 complex, DAXX, ASF1A-B, MCM2) demonstrated strong preference for WT histones, again, as one might expect from their structural characterisation with H3-H4 dimers (*Elsässer et al., 2012*; *English et al., 2005*; *Natsume et al., 2007*; *Huang et al., 2015*; *Richet et al., 2015*; *Wang et al., 2015*).

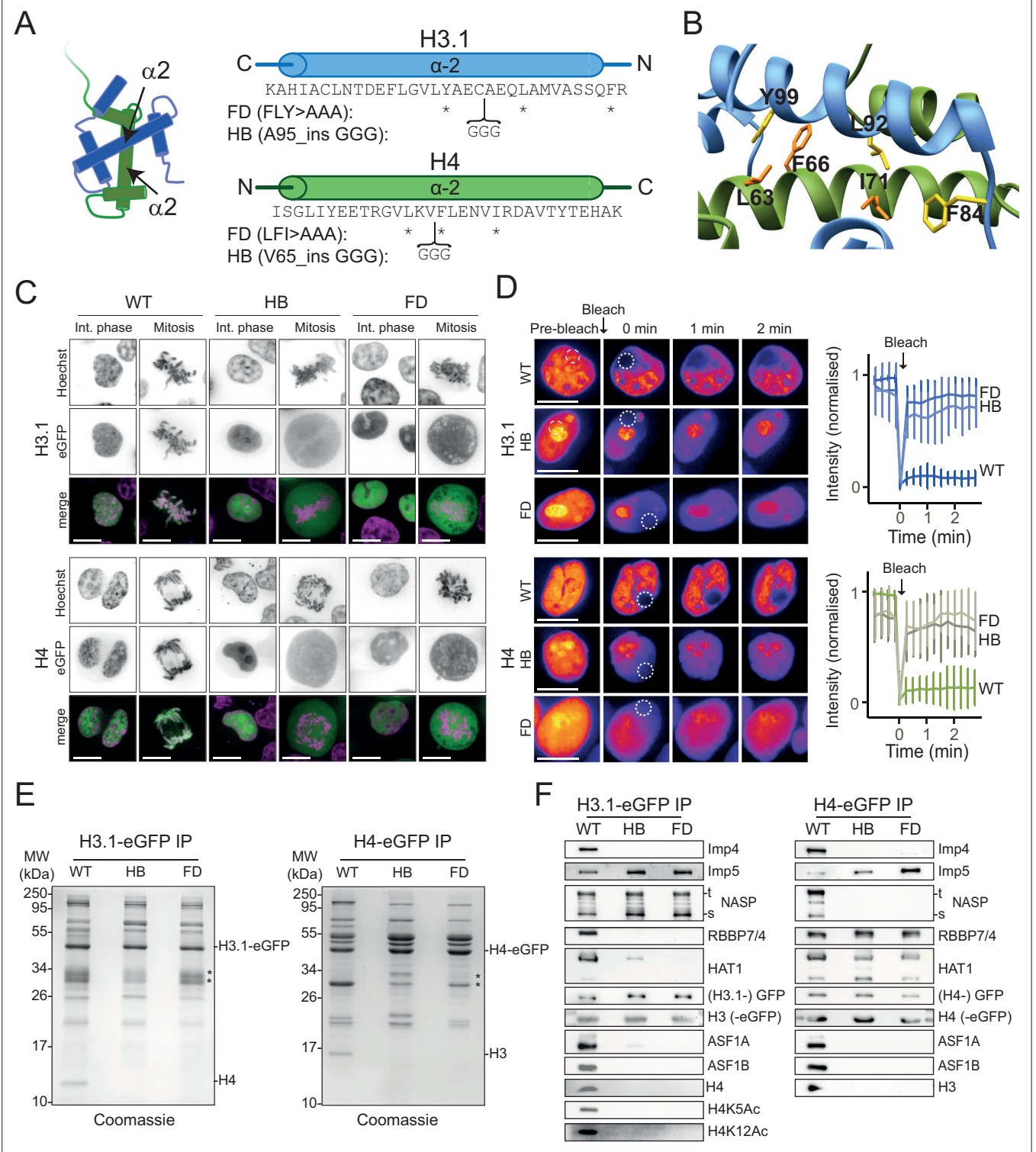

**Figure 1.** H3 and H4 dimerisation mutants translocate to the nucleus but do not incorporate into chromatin. (**A**) Positions of fold-disruptor (FD) substitutions (asterisks) and helix-breaker (HB) insertions (brackets). (**B**) Positions of the FD mutations superimposed on the H3-H4 heterodimer (PDB: 2HUE). Residues targeted are shown in yellow for H3 and in orange for H4. (**C**) Non-folding mutants were expressed as eGFP C-terminal fusions and transiently transfected in HEK293-F cells. Mutants localise to the nucleus similar to wild-type, but do not incorporate into chromatin as visualised by cytoplasmic fluorescence during mitosis. Scale bar indicates 10 μm. (**D**) Fluorescence recovery after photo-bleaching (FRAP) analysis of wild-type and mutant histones. Bleached regions of the nuclei at t = 0 min are indicated. Immediate recovery of fluorescence after bleaching shows that the histone mutants are not chromatin-bound as are their wild-type counterparts. Note that histone mutants have a tendency to accumulate in nucleoli.

*Figure 1 continued on next page*

*Figure 1 continued*

(**E**) Wild-type and mutant eGFP fusions were affinity-purified using an anti-GFP nanobody, separated by SD-PAGE and stained with Coomassie. Positions of tagged and endogenous histones are indicated. Asterisk indicates free GFP and partial degradation products. (**F**) Western blot analysis of immunoprecipitated wild-type and histone mutants as shown in (**E**) probed for known histone interactors.

The online version of this article includes the following figure supplement(s) for figure 1:

**Figure supplement 1.** Calibration for precise input quantification.

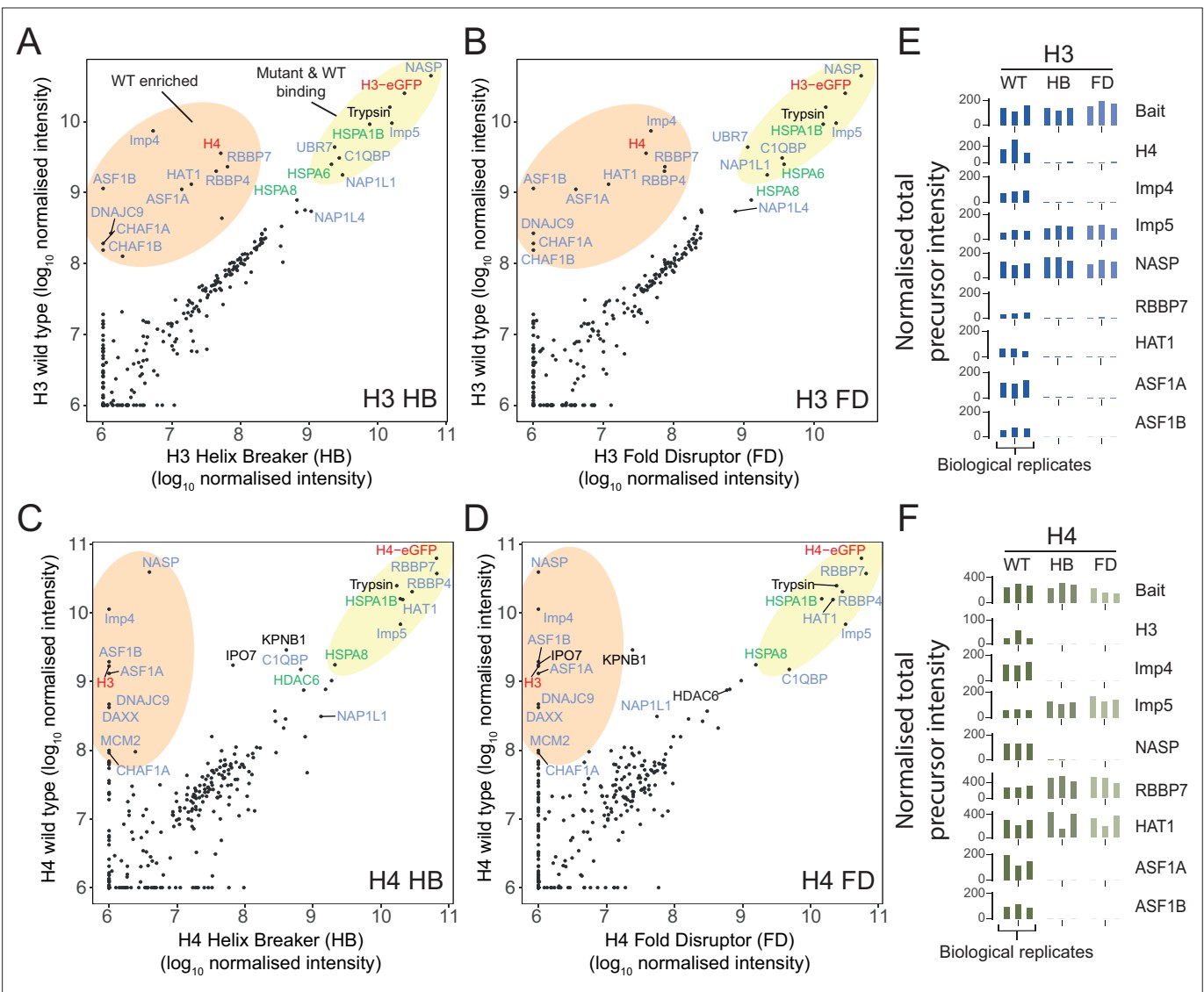

**Figure 2.** Proteomic analysis of wild-type (WT) and monomeric histone mutants. (**A–D**) Mutant versus WT scatter plots of normalised total precursor intensity from H3 vs. HB (**A**), H3 vs. FD (**B**), H4 vs. HB (**C**), and H4 vs. FD (**D**). Average of three experiments. By plotting mutants versus WT, factors that interact at similar levels will lie on the diagonal, with those that have preference for the dimer or monomer to fall above or below the diagonal, respectively. Circled regions show factors that are enriched for the WT (orange) or that are equally enriched on the WT and mutant (yellow). Previously reported histone-binding proteins are coloured in blue, with HSC70 family members coloured in green. Histones are coloured in red. (**E, F**) Alternative representation of proteomics data showing quantitative values (normalised total precursor intensity) on a linear scale ($1 \times 10^6$) for each biological replicate of key factors identified in (**A–D**). HB, helix-breaker mutation; FD, fold-disruptor mutation.

The online version of this article includes the following source data for figure 2:

**Source data 1.** Processed mass spectrometry data for CoIP of H3-eGFP WT and mutants 'helix breaker' (H3 HB) and 'fold-disruptor' (H3 FD).

**Source data 2.** Processed mass spectrometry data for CoIP of H4-eGFP WT and mutants 'helix breaker' (H4 HB) and 'fold-disruptor' (H4 FD).

Intriguingly, two recently discovered histone chaperones UBR7 (*Hogan et al., 2021*) and C1QBP (*Lin et al., 2021*) retained interactions with monomeric histones, with UBR7 showing specificity for histone H3 while C1QBP interacted with both H3 and H4. Furthermore, NAP1-like proteins NAP1L1 and NAP1L4 also bound to monomeric histones, albeit at much lower levels than other chaperones. Taken together, these results show that monomeric mutants of H3 and H4 retain distinct interaction profiles compared to WT and confirm Imp5, but not Imp4, as interacting with histones in their monomeric state.

## NASP forms discrete H3 monomer and H3-H4 dimer containing complexes

NASP can chaperone a monomer of H3 in vitro (*Bowman et al., 2017*), bind to an epitope obscured in the H3-H4 interface (*Bowman et al., 2016*; *Bao et al., 2022*), and pull down superstoichiometric amounts of H3 over H4 when isolated from HeLa cell extracts (*Apta-Smith et al., 2018*). To probe the NASP-associated histone pool at endogenous levels, we tagged NASP at its endogenous locus with TEV-cleavable (TEVcs) eGFP using CRISPR (*Figure 3A*). For comparison of H3-H4 dimer binding, we also knocked in a SpotTag-mCherry-3Ccs tag in frame with the first exon of ASF1B (*Figure 3A*, *Figure 3—figure supplements 1 and 2*), and purified these constructs using a single-step GFP-/RFP-Trap pulldown, alongside sNASP expressed transiently in HEK293-F cells (*Figure 3—figure supplement 1*). sNASP overexpression in HEK293-F cells and NASP expression from its endogenous locus in HeLa showed at least a twofold molar excess of H3 over H4, whereas ASF1 showed close to equimolar amounts in each case (*Figure 3B and C*), demonstrating that under endogenous conditions NASP binds to a superstoichiometric amount of H3 over H4.

To determine whether this excess of H3 relates to a sub-pool of NASP-H3 monomer, we performed biochemical fractionation of sNASP-containing complexes from HEK293-F cells (insufficient material prevented us from using endogenously tagged HeLa lines; however, the similar ratios of H3 to H4 in both lines suggest transient expression in HEK293 is representative of the endogenous scenario). Associated complexes were affinity captured using GFP-Trap resin before separation on either a glycerol gradient (*Figure 3D*) or by native PAGE (*Figure 3E*). SDS-PAGE analysis of gradient fractions revealed an sNASP-H3 monomer complex that could be unambiguously separated from other H3-H4 containing complexes (*Figure 3*, fractions 6–8, arrows). Western blotting confirmed the presence of sNASP and H3 in this complex, and the absence of H4, in addition to the absence of co-chaperones HAT1, RBBP7, ASF1A, and ASF1B, all of which eluted with the H3-H4 dimer displaying the H4K5Ac mark of newly synthesised histones (*Figure 3D*). Interestingly, UBR7 came down with sNASP, but did not migrate with the main sNASP-H3 species, as may be expected from our monomeric histone mutant analyses (*Figure 2A and B*), but overlapped with the broad sNASP co-chaperone peak (*Figure 3D*).

Native PAGE resulted in greater separation compared to glycerol gradient centrifugation, with individual bands representing distinct complexes being observed (*Figure 3E*). To identify components of each band, further separation in a second dimension under denaturing conditions was performed before staining with Coomassie or probing by Western blot (*Figure 3E*). Free sNASP is clearly visible as the major species in band 3, with partially degraded versions as minor species in bands 1 and 2. sNASP-bound H3, without H4, migrated as a separate species in band 5 (arrow), whereas H3-H4 and co-chaperones eluted in various species from bands 6–11 (*Figure 3E*).

Curiously, a minor, higher molecular weight band was detected in the anti-NASP Western blot under a long exposure, which runs at a size indicative of the tNASP isoform (*Figure 3E*). This was intriguing as only the shorter sNASP isoform was expressed as the eGFP-fusion and may relate to a low level of dimerisation with the endogenous protein, which has been suggested from the crystal structure (*Bao et al., 2022*; *Liu et al., 2021*) and biochemical analysis (*Campos et al., 2015*). Interestingly, only the s, and not the t, isoform co-migrated with the H3 monomer band. To investigate this further, we used an HEK293-F CRISPR knock-in cell line to separate complexes from both isoforms under endogenous expression levels (*Figure 3F*). As expected, the CRIPSR knock-in showed similar amounts of both splice isoforms, and again only the sNASP isoform co-migrated with an H3 monomer species (*Figure 3F*, arrow), suggesting that monomer binding may be an sNASP-specific function.

To more accurately assign components to each species, bands from sNASP 1D native gels were excised and subjected to mass spectrometry (*Figure 3G*). Label-free quantification (LFQ) of total precursor intensity for top hits (full list in *Figure 3—figure supplement 3* and *Figure 3—source*

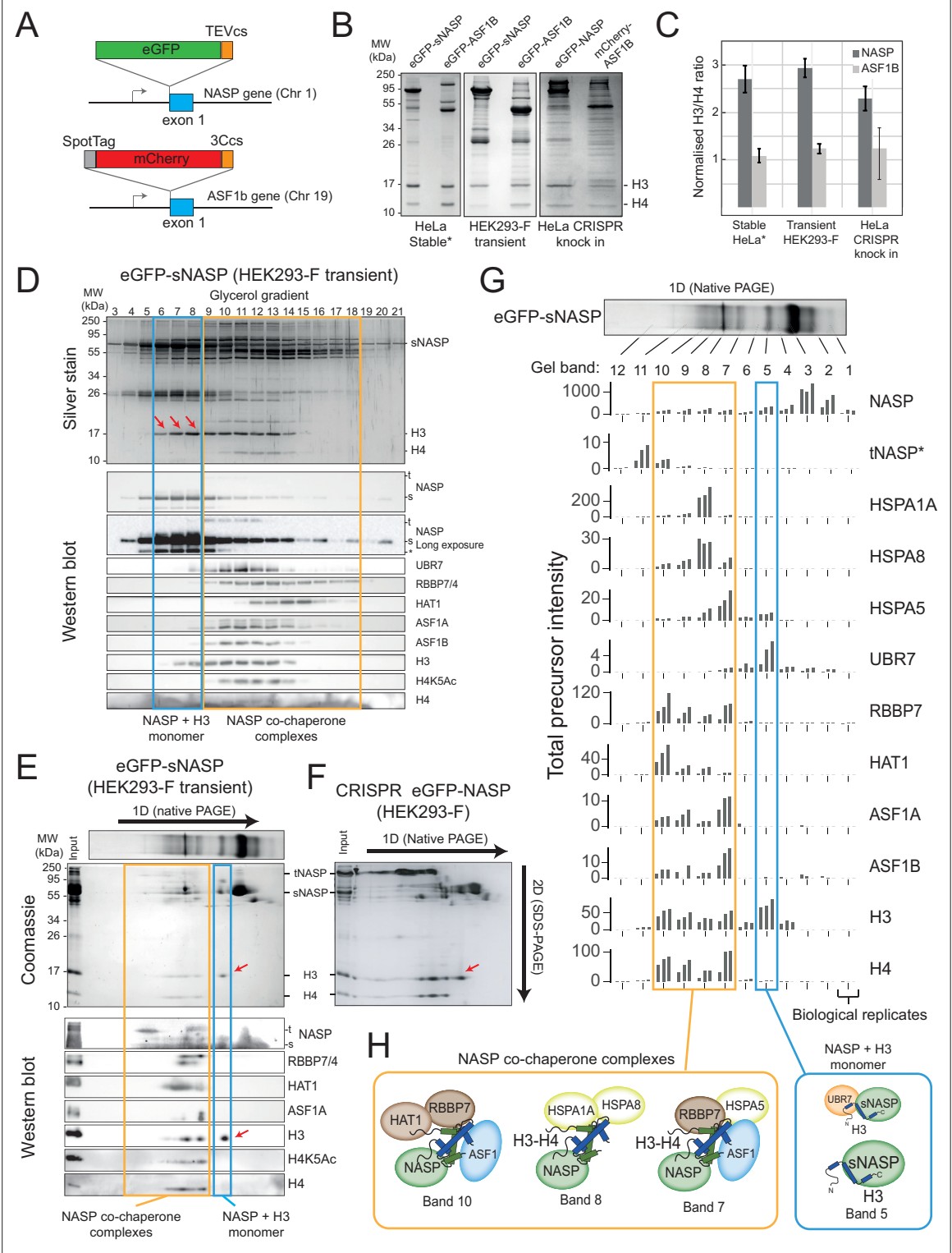

**Figure 3.** NASP associates with an H3 monomer and H3-H4 dimers in discrete, multi-chaperone complexes. (**A**) CRISPR knock-in tagging scheme. Endogenous NASP was tagged on exon 1 with eGFP-TEVcs and ASF1B with SpotTag-mCherry-3Ccs. (**B**) Histone chaperone immunoprecipitations separated by 15% SDS-PAGE and stained with Coomassie, for eGFP-sNASP and eGFP-ASF1B as stable HeLa cell lines, HEK293-F transient expression and CRISPR knock-in (as described in **A**). Asterisks mark previously published data (*Apta-Smith et al., 2018*). (**C**) Ratios of H3 compared to H4 associated with NASP and ASF1 from extracts stably expressed in a HeLa cell line, transiently expressed in HEK293-F cells or endogenously expressed in a CRISPR knock-in HeLa cell line. Background-corrected densitometry profiles for H3 and H4 were compared with purified recombinant H3-H4 dimers

*Figure 3 continued on next page*

Figure 3 continued

(n = 3, error bars = SD). (**D**) Fractionation by ultra-centrifugation on a glycerol gradient (4–20%) of immunoprecipitated eGFP-sNASP from transiently transfected HEK293-F cells (as in **B**) followed by SDS-PAGE, silver staining (top), and Western blot (bottom). Lane numbers correspond to glycerol fractions. Note that a species containing H3 but not H4 is clearly discernible (red arrows). (**E**) Native gel separation of immunoprecipitated eGFP-sNASP transiently expressed in HEK293-F cells (as in **B**) followed by SDS-PAGE, Coomassie staining (top) and Western blot (bottom). Blots were probed for known NASP-interacting factors. Note that a species containing H3 but not H4 is clearly discernible (red arrows). Band numbers correspond to gel sections that were analysed by mass spectrometry (as shown in **G**). (**F**) 2D native/SDS-PAGE as shown in (**E**), but using eGFP-NASP pulldowns from a HEK293-F knock-in cell population created identically to that shown in (**A**). Note that the tNASP isoform is more prevalent, but only the sNASP isoform co-migrates with monomeric H3. (**G**) Mass spectrometry identification of sNASP-interacting factors from gel slices shown in (**E**) (total precursor intensity). tNASP-specific peptides are shown separately to pan-NASP peptide. (**H**) Diagrammatic representation of NASP-containing complexes.

The online version of this article includes the following source data and figure supplement(s) for figure 3:

**Source data 1.** Processed mass spectrometry data for eGFP-sNASP CoIP native gel bands.

**Figure supplement 1.** Comparision of wild type NASP with transiently expressed eGFP-sNASP and quantification of H3-H4 staining.

**Figure supplement 2.** Validation of eGFP-TEVcs-NASP and mCherry-3C-ASF1b knock-in cell lines.

**Figure supplement 3.** Related to *Figure 3E and G*.

data 1) was plotted for each band in sequence, with the corresponding profiles revealing distinct sub-complexes associated with sNASP (*Figure 3G and H*). Band 5 was validated as containing monomeric H3 bound to sNASP without H4 and, surprisingly, UBR7. The UBR7 containing complex was estimated as roughly one-tenth of the NASP-H3 signal, which may explain the elution of UBR7 in the glycerol gradient: that is, the minor NASP-H3-UBR7 complex elutes separately from the major NASP-H3 complex on a glycerol gradient, but similarly in native PAGE. Band 7 contained peak signals for the HSC70 protein HSPA5, in addition to the histone chaperones RBBP7, ASF1A and B, and both H3 and H4, whereas band 8 contained peak signals for HSPA1A and HSPA8 together with a smaller proportion of H3 and H4. The NASP complex in band 9 contained the co-chaperones HAT1, RBBP7, and ASF1A and B, along with H3 and H4. DNAJC9 was not detected, suggesting that it does not form a complex with sNASP.

In summary, NASP forms distinct complexes containing either an H3 monomer or H3-H4 dimers, with monomeric H3 associating specifically with the s isoform. A sub-population of the NASP-H3 monomer pool is bound to the putative E3 ubiquitin ligase UBR7, previously shown to associate with H3 (*Campos et al., 2015*; *Kleiner et al., 2018*; *Lambert et al., 2015*) and NASP (*Hogan et al., 2021*; *Isobe et al., 2020*), whilst H3-H4 dimer-bound forms split into at least three complexes that differentially contain co-chaperones HAT1, RBBP7, ASF1A-B, and/or the HSC70 family proteins HSPA1A, HSPA5, and HSPA8. Strikingly, we did not detect Imp5 in NASP-bound complexes despite it co-purifying with monomeric H3 (*Figure 2D*), suggesting that NASP and Imp5 form mutually exclusive complexes with histones.

## Imp5 and sNASP are mutually exclusive in binding H3 and require RanGTP for hand-off

We next investigated the interplay between Imp5 and sNASP binding. To prevent extraneous factors affecting our analysis, we used purified recombinant proteins reconstituted in vitro. Imp5 was able to bind equimolar concentrations of H3 and H4 when separated by size-exclusion chromatography, demonstrating a direct interaction with histones, and interacted with both monomeric histones and H3-H4 dimers (*Figure 4—figure supplement 1*), as expected from its association with both WT and monomeric mutants (*Figure 2*).

As sNASP and Imp5 elution overlapped in size-exclusion chromatography (*Figure 4—figure supplement 1B*), we used glycerol gradient ultracentrifugation to separate the two species (*Figure 4A and B*). Purified Imp5-H3 complex was combined with an equimolar amount of sNASP and left to equilibrate for 3 hr on ice before separation. Interestingly, Imp5-H3 and sNASP eluted in separate fractions, suggesting direct competition for binding sites (*Figure 4B*). In the reverse experiment where Imp5 was added to the NASP-H3 complex and equilibrated for 3 hr on ice, Imp5 was able to strip sNASP of its histone, with almost all H3 co-fractionating with Imp5 (*Figure 4—figure supplement 1D*). This suggests that, firstly, as pre-assembled complex was used as input, NASP-H3 has a dissociation rate significantly below the incubation time for competition (3 hr), secondly, Imp5 has a significantly higher

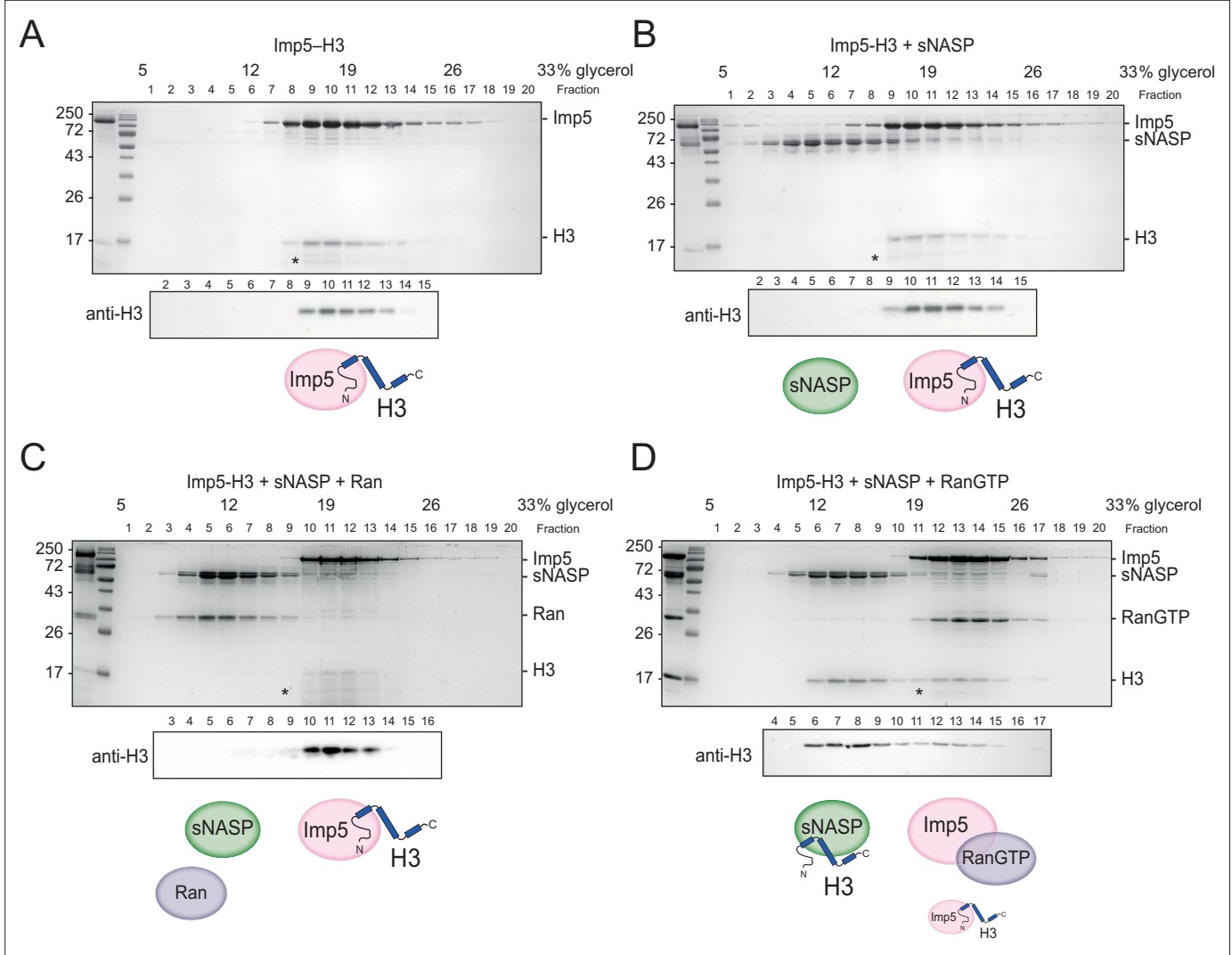

**Figure 4.** H3 binding by Imp5 and sNASP is mutually exclusive and relies on RanGTP for histone transfer. (**A**) Imp5-H3 reconstituted complex was purified by size-exclusion chromatography (SEC) and incubated on ice (control) for 3 hr before separating through ultracentrifugation on a 5–40% glycerol gradient. (**B**) As in (**A**), reconstituted complex was incubated on ice with equimolar concentration of sNASP (competition assay) for 3 hr before ultracentrifugation. H3 elutes with Imp5, whereas sNASP elutes in its separate fraction. (**C**) As in (**B**), but adding Ran in equimolar concentrations to sNASP and Imp-H3. Ran in its purified state is unable to bind to Imp5 and compete with H3. (**D**) As in (**B**), but adding RanGTP in equimolar concentrations to sNASP and Imp5-H3 complex. In this instance, RanGTP associates with Imp5 and displaces H3, which co-elutes with its chaperone sNASP. Asterisks indicate an Imp5 degradation product.

The online version of this article includes the following figure supplement(s) for figure 4:

**Figure supplement 1.** Related to *Figure 4*.

dissociation constant than NASP towards monomeric H3, and thirdly, the dissociation of Imp5-H3 in vivo likely requires additional factors, such as release by RanGTP, rather than being mediated by direct competition.

To investigate this further, we purified bacterially expressed Ran and added it at equimolar concentrations to sNASP and Imp5-H3, allowing it to equilibrate for 3 hr on ice. In its purified state, which most likely represents the GDP-bound form, Ran was unable to bind to Imp5 and displace H3. To favour the GTP-bound state, Ran was incubated with 5 mM EDTA and 50 mM GTP for 20 min on ice (*Schwoebel et al., 2002*), before performing the same competition experiment as above. Interestingly, RanGTP was bound to Imp5, displacing the majority of its H3 cargo, which was transferred to sNASP (*Figure 4C and D*).

In summary, direct binding competition towards H3 likely represents the absence of complex formation between Imp5 and NASP that we see in vivo, with histone transfer mediated by the canonical RanGTP pathway.

## Imp5 is the primary importin associated with H3 in the cytoplasm, whereas NASP binds only after nuclear import

Both Imp4 (*Tagami et al., 2004*; *Alvarez et al., 2011*; *Campos et al., 2015*) and Imp5 *Alvarez et al., 2011* have been shown to interact with H3-H4 in cytosolic extracts, with Imp4 being widely regarded as the primary import factor (*Campos et al., 2015*; *Tagami et al., 2004*; *Alvarez et al., 2011*; *Bernardes and Chook, 2020*). As cytoplasmic extracts often contain soluble nucleoplasmic proteins (*Paine et al., 1983*; *Paine et al., 1992*; *Apta-Smith et al., 2018*), we wanted to test interactions between H3, Imp4, and Imp5 in situ in living cells.

To do this, we first employed mito-F2H (*Bowman et al., 2016*) to probe their association using live-cell fluorescence microscopy. Mito-F2H builds on the principle of the F2H (fluorescence 2-hybrid) assay (*Zolghadr et al., 2008*), but uses the outer mitochondrial membrane (OMM) as a cytoplasmic tether rather than an integrated LacO array in the nucleus (*Figure 5A*). Using this technique, we previously demonstrated that tethered H3-eGFP was monomeric and associated with mCherry-Imp4 (*Apta-Smith et al., 2018*). To probe the interaction with Imp5, we co-expressed tethered eGFP with mCherry-Imp5 and quantified the colocalisation using Pearson's correlation coefficient (R). As expected, mCherry-Imp5 interacted strongly with tethered H3 at a level comparable to Imp4 (*Figure 5B and C*), suggesting that under conditions of overexpression both importins can interact with cytoplasmic H3.

One drawback of mito-F2H is the static snapshot it gives of factors in the cytoplasm, whereas probing interactions dynamically as histones translocate to the nucleus would be more informative. To address this, we used a previously developed pulse-chase method called 'RAPID-release' (Rapamycin Activated Protease through Induced Dimerisation and *release* of tethered cargo) (*Apta-Smith et al., 2018*). RAPID-release consists of two components: the histone cargo with a cytoplasmic tether localising to the outer mitochondrial membrane and a protease that can be locally activated through rapamycin-induced recruitment (*Apta-Smith et al., 2018*). The system is highly adaptable and can be modified to study both localisation and interactions using appropriate functional tags such as fluorescent proteins and/or proximity ligation tags, such as APEX2 (*Figure 5D*).

Firstly, to test the dynamic nature of Imp5's association with tethered H3, we released H3-mCherry in the presence of co-expressed eGFP-Imp5. The colocalisation observed under static conditions (*Figure 5C*, *Figure 5—source data 1*) disappeared when rapamycin was added to the cells, with H3 being imported rapidly and Imp5 remaining in the cytoplasm (*Figure 5E*, top), as quantified by the reduction in Pearson's R value (*Figure 5E*, bottom, *Figure 5—source data 2*), suggesting that the Imp5-H3 interaction is cytoplasm-specific, and that hand-off to the nuclear chaperoning machinery is a rapid event.

Secondly, to address which factors associate with tethered H3 proteome-wide, we incorporated an APEX2 tag into the tethered construct in a doxycycline-inducible FRT HeLa cell line. The APEX2 peroxidase allows for time-resolved, fast proximity biotinylation, requiring the xenobiotic biotin-phenol as a substrate and the addition of hydrogen peroxide for 1 min as catalyser (*Lam et al., 2015*). Preliminary analysis showed that the APEX fusion presented slower release kinetics when compared to H3-eGFP alone, possibly due to the increase in steric hindrance derived from a considerably larger fusion construct, with the majority of tethered H3 being released 60 min after addition of rapamycin, and all being released by 120 min (*Figure 5F*, *Figure 5—source data 3*). We performed proximity ligation reactions in triplicate at 0 min (cytoplasmic interactions), 10 min (mixed), and 120 min (nuclear interactions) after rapamycin addition, allowing a 1 min labelling window. After quenching, biotinylated peptides were isolated under denaturing conditions. In this way, ideally only interactions that occurred in the cell and not during post-lysis procedures were recorded.

The 0 min and 10 min datasets were very similar, owing to cleavage of the H3-eGFP-APEX2 tag being slower than the APEX2-free version (*Figure 5F*, *Figure 5—figure supplement 1*), restricting our analysis to 0 and 120 min, relating to cytoplasmic and nuclear interactions, respectively. To control for unspecific interactions, we filtered our results against a control comprised of tethered APEX without H3. Enrichment in the nucleus or cytoplasm was visualised in a volcano plot by plotting the LFQ ratio against a probability value derived from three biological replicates (*Figure 5G*, *Figure 5—source data*

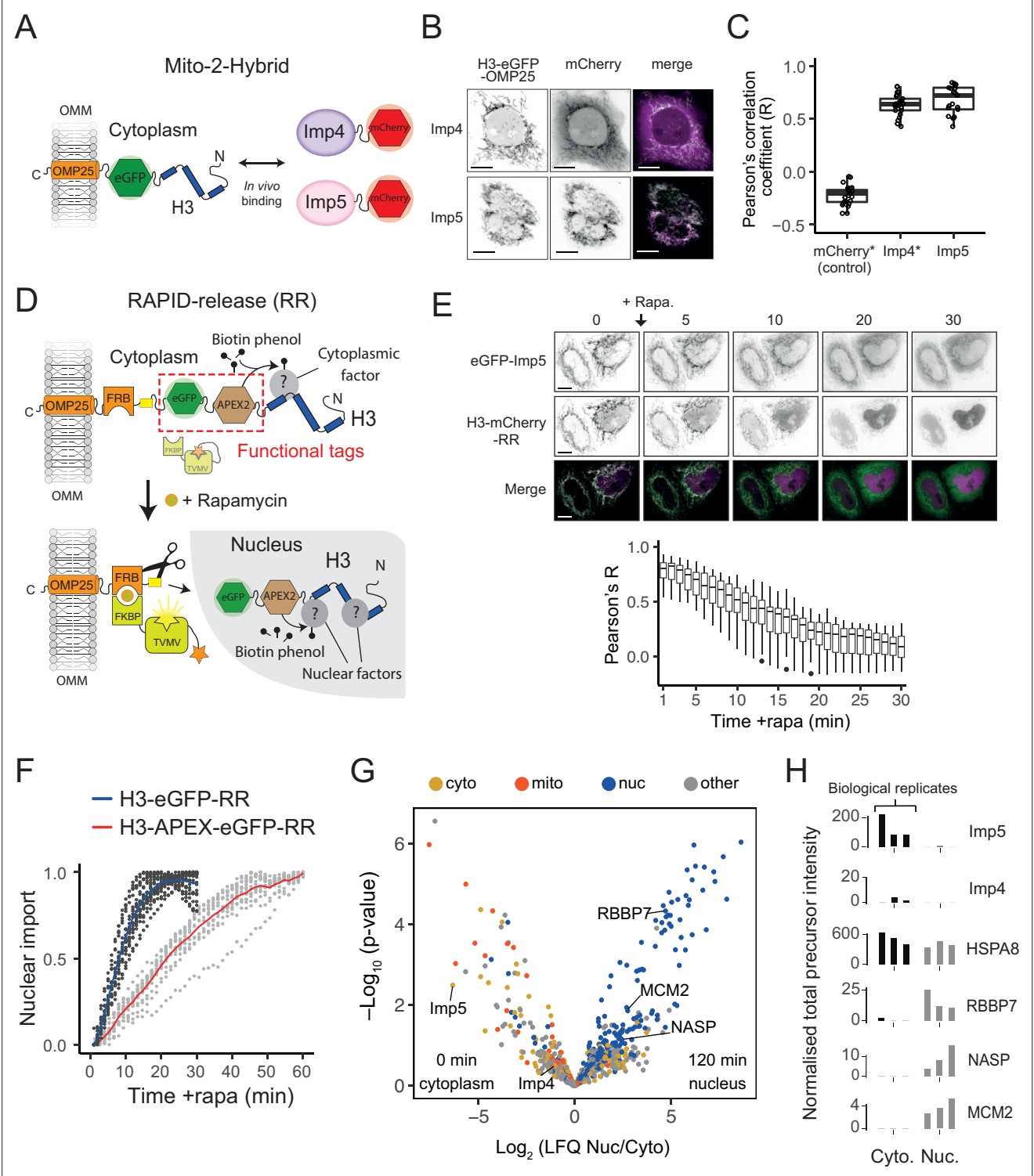

**Figure 5.** Imp5 is the predominant importin bound to nascent H3 in the cytoplasm, with histone chaperones binding only after nuclear import. (**A**) In the Mito-2-Hybrid assay, cytoplasmic interactions are detected by tethering of a bait protein (H3) to the outer mitochondrial membrane (OMM) via an OMP25 tail-anchoring peptide. Recruitment of a co-expressed soluble protein reports an interaction with the bait. Scale bar = 10 μm. (**B**) Representative images of Imp4 and Imp5 interaction with tethered H3. (**C**) Quantification of interactions shown in (**B**) using the Pearson's correlation coefficient as a measure of colocalisation. Asterisks indicate plotting of previously published values (*Apta-Smith et al., 2018*). (**D**) Experimental design for incorporating RAPID-Release (RR) into the cytoplasmically tethered H3 construct. Functionalisation is introduced by adding required tags to the tether (top, examples in this case being eGFP and APEX2), which can be released along with the cargo (H3) to study interaction dynamics after the addition of

*Figure 5 continued on next page*

*Figure 5 continued*

rapamycin (bottom). (**E**) Imp5 dissociates from H3 as it is released from its tether and enters the nucleus, suggesting a rapid hand-over mechanism. (**F**) Nuclear import rate of H3.1-eGFP and H3.1-APEX-eGFP. Single cells are represented as grey dots. Averages are highlighted in blue (H3.1-eGFP) and red (H3.1-APEX-eGFP). Note that H3.1-eGFP nuclear import plateaus within 20 min, whereas H3.1-APEX-eGFP requires up to 60 min. (**G**) Volcano plot representing label-free mass spectrometry quantification analysis (LFQ) after streptavidin pulldown of proximity labelled proteins at 0 min (cytoplasm) or 120 min (nucleus) after the addition of rapamycin. Interacting factors are colour-coded according to their main subcellular location: cytoplasm (cyto) yellow, mitochondria (mito) orange, nucleus (nuc) blue, and other in grey (The Human Protein Atlas version 21.1). (**H**) Alternative representation of proteomics data showing quantitative values (normalised total precursor intensity) on a linear scale ($1 \times 10^6$) for each biological replicate of key factors identified in (**F**).

The online version of this article includes the following source data and figure supplement(s) for figure 5:

**Source data 1.** mCherry (control), mCherry-Imp4, and mCherry-Imp5 versus H3-eGFP-OMP25 Pearson's correlation coefficients used in *Figure 5C*.

**Source data 2.** eGFP-Imp5 and H3-mCherry-RAPID-release Pearson's correlation coefficients used in *Figure 5E*.

**Source data 3.** H3.1-eGFP-RAPID-release and H3.1-APEX2-eGFP-RAPID-release nuclear localisation used in *Figure 5F*.

**Source data 4.** Processed mass spectrometry data for APEX2-OMP25 control.

**Source data 5.** Processed mass spectrometry data for H3.1-APEX2 at 0 and 120 min after addition of rapamycin.

**Figure supplement 1.** Validation of the FRT-H3-APEX2-eGFP-RR cell line.

*4 and 5*), with the triplicate LFQ scores for specific hits plotted individually alongside (*Figure 5H*). Colouring hits based on their nuclear localisation, as determined by the Human Protein Atlas (https://www.proteinatlas.org/), revealed annotated cytoplasmic and mitochondrial proteins enriched at 0 min, and annotated nuclear proteins present at 120 min, as expected (*Figure 5G*).

Both Imp4 and Imp5 were detected bound to H3 in the cytoplasm, but Imp5 was ~100-fold enriched over Imp4 (*Figure 5G and H*), demonstrating that at its endogenous level Imp5 is the primary importin associated with cytoplasmically tethered H3. NASP was only detectable after H3 had translocated to the nucleus, as were other known histone chaperones such as RBBP7, RBBP4 (RbAp48), and MCM2 (*Figure 5G and H* and *Figure 5G and H*, *Figure 5—source data 5*), supporting our previous findings that NASP interacts with histones in the nucleus and not in the cytoplasm (*Apta-Smith et al., 2018*). Interestingly, we detected the HSC70 family member HSPA8, which bound to H3 comparably across the cytoplasm and the nucleus (*Figure 5G and H*), suggesting a compartment-independent role for this chaperone. Histone H4, as a common contaminant, was identified at similar levels across samples including the tethered APEX control, suggesting affinity for the tag or resin, and had to be excluded from analysis. In summary, in situ methods identify Imp5 as the major H3 binding importin in vivo, with NASP binding H3 only once it has translocated to the nucleus.

## NASP binds rapidly to newly imported H3

As the APEX2-tag suppressed the cleavage rate of tethered H3 in the RAPID-release assay, we were unable to probe early events following H3 import. To address this, we removed APEX and introduced the RAPID-release system into an eGFP-NASP CRIPSR cell line, swapping the H3-eGFP for H3-mCherry (*Figure 6A*). Endogenously expressed eGFP-NASP was entirely nuclear, with no enrichment over the tethered histone, further showing they do not interact in the cytoplasm (*Figure 6B*). Release of H3-mCherry by addition of rapamycin followed similar kinetics to previously described H3-EGFP (*Apta-Smith et al., 2018*), resulting in a half-maximum cleavage at around 10 min (*Figure 6—figure supplement 1*). Unlike Imp5 (*Figure 5E*), association could not be inferred from the microscopy alone as H3-mCherry is diffuse in the nucleus after release, similar to NASP. To address this, we performed co-IPs at 10 min, 1 hr, and 24 hr intervals after histone release (*Figure 6C*) to capture the interaction. Interestingly, H3's association with NASP peaked at the 10 min time point, when the nucleus experiences the highest surge of the released histone (*Figure 6C*, red arrows), strongly suggesting that NASP associates with H3 as it enters the nucleus. Association with H3 was also present at later time points, with 24 hr likely to represent an equilibrium between soluble and chromatin-bound state. This suggests that NASP can also interact with post-nucleosomal histone and is not necessarily limited to the interaction with incoming, newly synthesised H3. Taken together, these findings suggest that NASP is the predominant nuclear receptor for H3 monomers via a rapid hand-off event from Imp5.

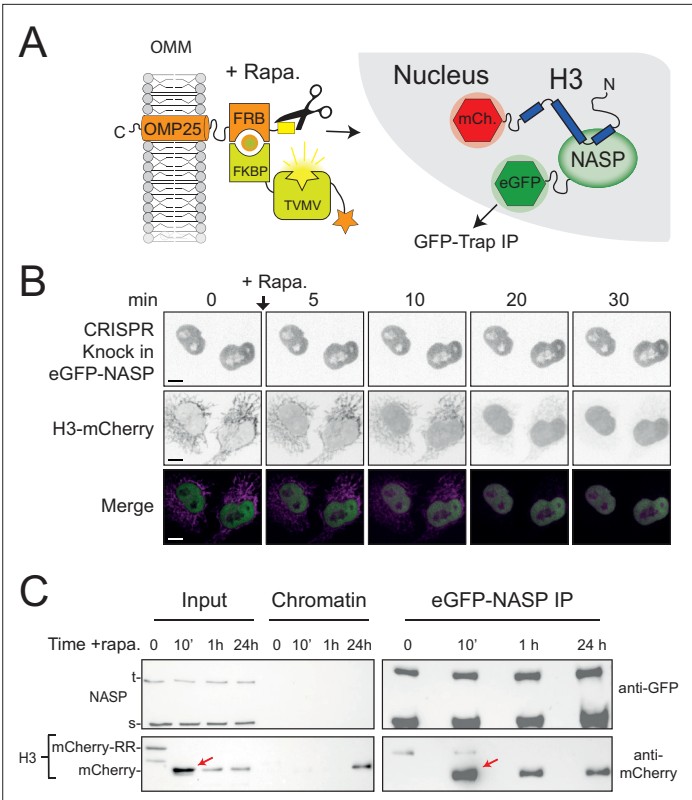

**Figure 6.** NASP binds rapidly to newly imported H3. (**A**) Experimental design combining a histone pulse with co-immunoprecipitation. Addition of rapamycin recruits an otherwise autoinhibited TVMV protease which cleaves the tethered histone fusion, allowing it to enter the histone deposition pathway. (**B**) Time course of H3-mCherry release in the CRISPR eGFP-NASP knock-in cell line. Note that endogenously tagged NASP does not associate with H3 tethered in the cytoplasm. (**C**) Western blot analysis of co-immunoprecipitated H3-mCherry from the cell line shown in (**B**) at 0, 10 min, 1 hr, and 24 hr time points after the addition of rapamycin.

The online version of this article includes the following figure supplement(s) for figure 6:

**Figure supplement 1.** Kinetics of H3 RAPID-release and co-localistation analysis with Imp5.

## Discussion

Rapid nuclear import and folding of histones is crucial for timely nucleosome deposition, functionalisation, and DNA safe-keeping. Several studies have identified Imp4 (*Tagami et al., 2004*; *Jasencakova et al., 2010*; *Campos et al., 2010*; *Alvarez et al., 2011*; *Ask et al., 2012*; *Campos et al., 2015*), but also Imp5 (*Mühlhäusser et al., 2001*; *Alvarez et al., 2011*), as interacting with histones H3 and H4. In this study, we provide evidence that Imp5 is the primary importin bound to H3 when detected by in situ methods, and that, in contrast to Imp4, Imp5 interacts preferentially with a monomer of H3.

Due to the lack of structural information regarding Imp5's interaction with histones, it is not immediately clear why there is a functional difference in their binding. A recent structure of Imp4 in complex with H3-H4 and ASF1 (*Bernardes et al., 2022*) agrees with our finding that Imp4 does not interact with monomeric H3 and H4 mutants, in that, although Imp4 interacts predominantly with the N-terminal region of H3, it also makes contacts with the histone fold domain of H4. This is in contrast to Imp9's interaction with its H2A-H2B cargo, where it fully encapsulates an H2A-H2B dimer (*Padavannil et al., 2019*). Thus, whilst a divergent histone interaction between Imp4 and Imp5 should not be ruled out, it seems likely that Imp5 binds to H3 and H4 in a similar way as Imp4, but potentially with more emphasis on the H3 N-terminal region, having lost the necessity for an H3-H4 dimer substrate.

Imp5 binding to H3 precludes NASP interaction, both in proteomic analyses and in in vitro binding experiments, with the nuclear localisation of NASP suggesting NASP acts downstream of Imp5 in the histone chaperoning pathway. The TPR domain of NASP has two distinct interaction sites with histone H3: an H3 N-terminal binding surface and an H3α3-binding pocket (*Bao et al., 2022*; *Liu et al., 2021*;

*Bowman et al., 2016*; *Bowman et al., 2017*). Overlapping binding sites likely explains the H3 N-terminal competition between importin and chaperone that we see; however, if this were the only Imp5 interaction site, one might expect NASP to remain bound through the H3α3 (*Bowman et al., 2016*; *Bao et al., 2022*). Additional interactions between Imp5 and the histone fold domain of H3 could be one explanation for the lack of trimeric complex. Alternatively, H3α3 binding by NASP may be more complex and require additional factors. In either case, we show it is RanGTP that provides the switch necessary for release of H3 from the importin, which is in contrast to Imp9-H2A-H2B, where RanGTP alone was insufficient (*Padavannil et al., 2019*), highlighting further diversity amongst importin-β proteins and their handling of histones.

If newly synthesised H3 and H4 are rapidly imported into the nucleus as monomers bound to Imp5, what is the role of the long-established Imp4-H3-H4-ASF1 complex? The most likely explanation would be a parallel pathway for importing H3 and H4 that have folded in the cytoplasm (*Campos and Reinberg, 2010*; *Alvarez et al., 2011*). An alternative scenario could be that Imp4 recognises post-nuclear H3-H4-ASF1, promoting the re-import of H3-H4 dimers that may have escaped. Indeed, the H3-H4-ASF1 complex would be small enough to transit the nuclear pore passively (*Frey et al., 2018*), with ASF1, unlike NASP, not containing a nuclear localisation sequence.

The necessity to sequester monomeric histones in the nucleus may relate to the folding kinetics of H3-H4. In vitro studies suggest that H3-H4 folding is the rate-limiting step in the dimerisation pathway (*Banks and Gloss, 2004*), with a millisecond timescale comparable to that of nuclear pore transit (*Yang et al., 2004*; *Kubitscheck et al., 2005*). NASP may therefore be regarded as having a 'holdase' function, as has been suggested for the ATP-independent small heat shock protein family of molecular chaperones (*Jakob et al., 1993*; *Mymrikov et al., 2017*; *Ecroyd, 2015*; *Taipale et al., 2014*; *Mogk and Bukau, 2017*; *Reinle et al., 2022*), preventing aggregation of H3 in the nuclear milieu before encountering H4. It is interesting that it is specifically sNASP rather than tNASP that co-separates with the H3 monomer, suggesting alternative functions for these two splice isoforms. tNASP incorporates exon 8 that functionally extends the acidic region of sNASP. How this insertion mediates the differing binding profiles we see would be an interesting future question. The role of NASP as a holdase is easily reconciled with the idea of NASP overseeing a reservoir of soluble histones (*Cook et al., 2011*; *Hormazabal et al., 2022*), which would be compatible with it acting in the nucleus. In addition, the presence of NASP in downstream complexes (*Tagami et al., 2004*; *Groth et al., 2011*), mediated by interactions outside of the TPR-H3 peptide interface (*Bowman et al., 2017*; *Bao et al., 2022*; *Liu et al., 2021*), supports NASP guiding the transition from monomeric import to a heterodimer capable of DNA deposition (*Bowman et al., 2017*).

The holoenzyme HAT1-RBBP7's interaction with monomeric H4 mutants suggests that HAT1-RBBP7 may function as a counterpart to NASP-H3, accepting H4 from Imp5 in the nucleus prior to folding. In support of this, structural characterisation of the homologous Hat1p-Hat2p complex from yeast suggests that dimerisation with H3 is not necessary for interaction with H4, which is mediated predominantly with the H4 tail and α1 region (*Li et al., 2014*). The HAT1-RBBP7 complex is responsible for H4K5Ac and H4K12Ac on newly synthesised histone H4 (*Verreault et al., 1998*). Interestingly, in NASP pulldowns we find histone H4K5Ac peaking in bands with sparse HAT1, suggesting a catalytic turnover of HAT1, but with retention of RBBP7, which may be related to its interactions with H3 N-terminal tail (*Yue et al., 2022*). The low levels of ASF1 detected in complex with HAT1 are notable and suggest a transitory state, potentially involving competition between the TPR domain of NASP and ASF1 for the H3α3 helix (*English et al., 2005*; *Natsume et al., 2007*; *Bao et al., 2022*).

Unexpectedly, we found the ubiquitin ligase UBR7 co-migrating with a NASP-H3 monomer peak. UBR7 associates with H3 and its centromeric homolog CENP-A (*Campos et al., 2015*; *Foltz et al., 2009*; *Lambert et al., 2015*; *Kleiner et al., 2018*), with proposed interactions occurring between its PHD finger domain and the methylated tail of H3 (*Kleiner et al., 2018*). Additionally, UBR7 is also responsible for H2BK120-directed monoubiquitin ligase activity (*Adhikary et al., 2019*). The function of UBR7 in the context of soluble H3 is less clear and need not relate directly to the major function of the NASP-H3. Indeed, UBR7 co-migrated with sNASP and H3 in native PAGE, but did not fractionate with sNASP-H3 after glycerol gradient centrifugation, suggesting a distinct subcomplex population (*Figure 3*). According to our label-free quantification, UBR7 was roughly one order of magnitude lower than the NASP-H3 pool, further suggesting that it may have a secondary role in histone metabolism,

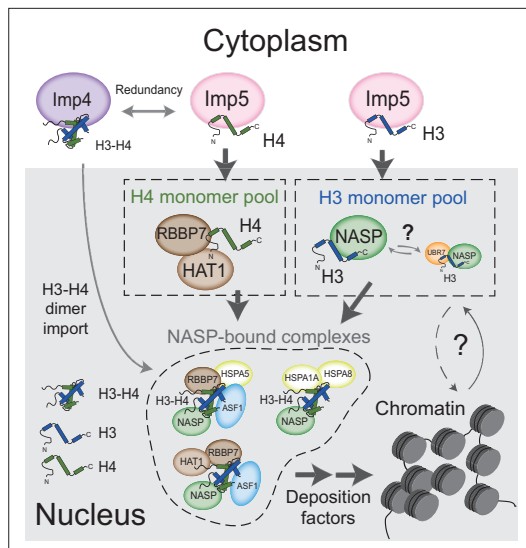

**Figure 7.** Model for nuclear translocation of histones H3 and H4 as monomers and processing in the nucleus Imp5 imports histone H3 and H4 as monomers whilst Imp4 has a preference for dimers, albeit with some redundancy. Upon nuclear translocation, RanGTP induces cargo dissociation with H3 being transferred to sNASP and H4 to the HAT1 complex. These chaperones may have a holdase function that buffers imbalances in individual histone supply. H3 and H4 fold and enter the deposition pathway via a number of distinct NASP-bound complexes.

potentially related to histone degradation (*Cook et al., 2011*; *Hormazabal et al., 2022*) or reposition (*Hogan et al., 2021*).

With regards to integration with the heat shock protein folding machinery, we observe interactions with different HSC70-type chaperones, namely, HSPA8, HSPA1A-B, and HSPA5. Intriguingly, HSPA8 forms a complex with NASP and H3-H4 which is distinct from other co-chaperoning complexes, perhaps relating to the degradation of misfolded histones, as previously proposed (*Cook et al., 2011*; *Hormazabal et al., 2022*). Their cross-compartment binding could also relate to the folding or re-folding of aggregated histones, as recently suggested for the HSP70 recruitment factor DNAJC9 (*Hammond et al., 2021*). We did not detect DNAJC9 associating in a co-chaperoning complex with NASP, which, along with DNAJC9's recognition of an H3-H4 dimer, may suggest that DNAJC9 acts after NASP binding or in a parallel pathway for aggregated or misfolded dimers. The presence of HSPA8 bound to cytoplasmically tethered H3 is in agreement with previous findings that show HSP70 proteins bind to newly synthesised H3 before folding with H4 (*Alvarez et al., 2011*; *Campos and Reinberg, 2010*). We also detect the new histone chaperone C1QBP (*Lin et al., 2021*) bound to monomeric mutants of both H3 and H4. However, we do not see association with NASP, suggesting a potential upstream function of this protein. The role of these novel histone-binding factors in the histone chaperoning pathway will require further analyses to reveal their function in detail.

In summary, we propose a model in which monomeric H3 and H4 are imported into the nucleus via an Imp5-dependent pathway (*Figure 7*). In this model, H3 and H4 are transferred to NASP and HAT1-RBBP7, respectively, dimerising in the nucleus rather than the cytoplasm. The ability to form stable nuclear pools of monomeric histones would provide an additional level of contingency for counteracting imbalances in histone supply, enabling a soluble reservoir of histone to be maintained under adverse cellular conditions.

## Limitations of the study

This study introduces new approaches to address the transient nature of soluble histones. The non-dimerising mutants greatly increase the pool of soluble histones for analysis, but could lead to non-physiological interactions. This does not seem to be an issue in this study as both mutants and wild type have overlapping interactomes, with mutants losing binding partners relative to the wild type rather than gaining. RAPID-release combined with APEX2 allows labelling in living cells in intact compartments (*Lam et al., 2015*). However, it produces higher concentrations of cytoplasmic H3 in the proximity to the OMM than would naturally occur, which may drive non-specific interactions to some degree. In addition, proximity-dependent biotinylation protocols suffer from promiscuous labelling biased towards ubiquitous factors, as well as peptide masking by naturally biotinylated proteins, potentially limiting interactome analysis (*Gingras et al., 2019*; *Kim and Roux, 2016*). Our results with H4 non-dimerising mutants imply that the HAT1 complex functions in parallel to sNASP-H3. However, an H4-HAT1 complex has yet to be isolated from cell extracts. Whether this is due to a transient or weaker interaction that does not survive biochemical extraction, or whether a monomeric H4 population represents the limiting factor for histone H3-H4 dimerisation, is still to be determined.

# Materials and methods

## Key resources table

| Reagent type (species) or resource | Designation | Source or reference | Identifiers | Additional information |
|---|---|---|---|---|
| Bacterial cells DH5-alpha (*Escherichia coli*) | DH5-alpha | Thermo | EC0112 | Chemically competent cells |
| Bacterial cells BL21 (DE3) (*Escherichia coli*) | BL21(DE3) | Thermo | EC0114 | Chemically competent cells |
| HeLa Kyoto cell line (*Homo-sapiens*) | Cancer cell line | Originally sourced from CRUK Cell Services | RRID:CVCL_1922 | |
| HeLa FRT cell line (*Homo-sapiens*) | T-REx-HeLa Cell Line | Invitrogen | R71407 RRID: CVCL_D587 | |
| HEK293-F cells | FreeStyle 293 F Cells | Invitrogen | R79007 RRID: CVCL_D603 | |
| Plasmid for transfected construct (human) | H3.1-EGFP | Backbone plasmid: pEGFP-N1 | *Apta-Smith et al., 2018* | transfected construct (human) |
| Plasmid for transfected construct (human) | H3.1 (A95_GGG)-EGFP | Backbone plasmid: pEGFP-N1 | This study | transfected construct (human) |
| Plasmid for transfected construct (human) | H3.1 (FLY >AAA)-EGFP | Backbone plasmid: pEGFP-N1 | This study | transfected construct (human) |
| | | | *Apta-Smith et al., 2018* | |
| Plasmid for transfected construct (human) | H4-EGFP | Backbone plasmid: pEGFP-N1 | | transfected construct (human) |
| Plasmid for transfected construct (human) | H4 (V65GGG)-EGFP | Backbone plasmid: pEGFP-N1 | This study | transfected construct (human) |
| Plasmid for transfected construct (human) | H4 (FLI >AAA)-EGFP | Backbone plasmid: pEGFP-N1 | This study | transfected construct (human) |
| Plasmid for transfected construct (human) | H3.2-mCherry-2xTVMVcs-FRB-OMP5–IRES–FKBP-TVMV-AI | pEGFP-C1 | This study | transfected construct (human) |
| Plasmid for transfected construct (human) | EGFP-Imp5 | pEGFP-C1 | This study | transfected construct (human) |
| Plasmid for transfected construct (human) | EGFP-TEV-sNASP | pIRESpuro2 | This study | transfected construct (human) |
| Plasmid for transfected construct (human) | H3.1-Flag-APEX2-EGFP-2xTVMVcs-FRB-OMP25-IRES-FKBP-TVMV-AI | pcDNA5/FRT/TO | This study | transfected construct (human) |
| Plasmid for transfected construct (human) | pNASP-EGFP-TEV_HDR-donor | pBlueScript II KS (+) | This study | transfected construct (human) |
| Plasmid for transfected construct (human) | pX461-NASP-sgRNA2-down | pX461-PSPCas9N(BB)–2A-GFP | This study | transfected construct (human) |
| Plasmid for transfected construct (human) | pX461-NASP-sgRNA2-up | pX461-PSPCas9N(BB)–2A-GFP | This study | transfected construct (human) |
| Plasmid for transfected construct (human) | pASF1b-Spot-mCherry-3C_HDR-donor | pBlueScript II KS (+) | This study | transfected construct (human) |
| Plasmid for transfected construct (human) | pX461-ASF1B-sgRNA2-down | pX461-PSPCas9N(BB)–2A-GFP | This study | transfected construct (human) |
| Plasmid for transfected construct (human) | pX461-ASF1B-sgRNA2-up | pX461-PSPCas9N(BB)–2A-GFP | This study | transfected construct (human) |
| Plasmid for transformation (human) | pGST-HRV 3Ccs-Imp5 iso1 | pGEX-6P1 | This study | Bacterial expression construct (human) |
| Plasmid for transformation (human) | pHis-TEVcs-Ran | pETMCN6His | This study | Bacterial expression construct (human) |

*Continued on next page*

*Continued*

| Reagent type (species) or resource | Designation | Source or reference | Identifiers | Additional information |
|---|---|---|---|---|
| antibody | Anti-ASF1A (C6E10), Rabbit monoclonal | Cell Signaling Technology | C6E10 | WB (1:1000) |
| antibody | Anti-ASF1B, Rabbit monoclonal | Cell Signaling Technology | C70E2 | WB (1:1000) |
| antibody | Anti-HAT1, Rabbit monoclonal | Abcam | ab194296 | WB (1:1000) |
| antibody | Anti-Histone H3, Rabbit polyclonal, IgG | Cell Signaling Technology | #9715 | WB (1:1000) |
| antibody | Anti-Histone H4, Rabbit polyclonal | Abcam | ab7311 | WB (1:500) |
| antibody | Anti-Histone H4 k12Ac, Rabbit polyclonal | Millipore (Merck) | #07–595 | WB (1:1000) |
| antibody | Anti-Histone H4 k5Ac, Rabbit polyclonal | Millipore (Merck) | #07–327 | WB (1:2000-1:5000) |
| antibody | Anti-Importin4, Rabbit monoclonal [EPR13660-27] | Abcam | ab181037 | WB (1:10000) |
| antibody | Anti-Karyopherin beta 3 (IPO5), Mouse polyclonal | Abcam | ab88695 | WB (1:500) |
| antibody | Anti-NASP, rabbit polyclonal | Home-made | NA | WB (1:20000) |
| antibody | Anti-mCherry, Rabbit polyclonal | Abcam | ab167453 | WB (1:5000) |
| antibody | Anti-RbAp46/48, Rabbit - monoclonal (IgG) | Cell Signaling Technology | (D4F8) #9067 | WB (1:1000) |
| antibody | Anti-UBR7, Rabbit polyclonal | Abcam | ab241371 | WB (1:1000) |
| antibody | GFP (B-2) HRP, Mouse - monoclonal (clone B-2), IgG2a | Santa Cruz | sc-9996 HRP | WB (1:1000 to 1:5000) |
| antibody | Goat Anti-Mouse HRP, Goat polyclonal, IgG | Abcam | ab205719 | WB (1:5000) |
| antibody | Anti-Rabbit, HRP-linked antibody, Goat polyclonal, IgG | Cell Signaling Technology | 7074 S | WB (1:10000 to 1:20000) |
| antibody | Goat Anti-Rabbit Alexa 568, Goat polyclonal, IgG | Abcam | ab175471 | WB (1:1000) |
| sequence-based reagent | XhoI_IPO5_F | This paper | PCR primers | aaaaaaCTCGAGcaA TGGCGGCGGCCGC |
| sequence-based reagent | KpnI_IPO5_R | This paper | PCR primers | GGTGGTggtaccTCAC GCAGAGTTCAGGAGCTC |
| sequence-based reagent | IPO5_pGEX_F | This paper | PCR primers | ctgttccaggggcccctggg atccGCAATGGC GGCGGCtGCG |
| sequence-based reagent | IPO5_pGEX_R | This paper | PCR primers | gtcagtcagtcacgatgcggccgc TCACGCAGA GTTCAGGAGC |
| sequence-based reagent | Kozak5p_F | This paper | PCR primers | TGAACCGTCAG ATCCGCTAGC |

*Continued on next page*

*Continued*

| Reagent type (species) or resource | Designation | Source or reference | Identifiers | Additional information |
|---|---|---|---|---|
| sequence-based reagent | OMP25_IRES_R | This paper | PCR primers | CGGTAGCGCTAC AGCTGTTTGCGATAGCG |
| sequence-based reagent | OMP25_IRES_F | This paper | PCR primers | TCGCAAACAGCTG TAGCGCTACCGGACTCAG |
| sequence-based reagent | IRES_FKBP_R | This paper | PCR primers | TCCACCTGCACCAT GGTTGTGGCCATAT |
| sequence-based reagent | FKBP_IRES_F | This paper | PCR primers | ATGGCCACAACCA TGGTGCAGGTGGAAACC |
| sequence-based reagent | FKBP_FRT_R | This paper | PCR primers | TGTGGGAGGTTTC TAGCTGCCCGGCGC |
| sequence-based reagent | FRT_AI_F | This paper | PCR primers | CCGGGCAGCTAGA AACCTCCCAC ACCTCCCCCT |
| sequence-based reagent | Kozak_H3.1_R | This paper | PCR primers | GATCTGACGG TTCACTAAAC |
| sequence-based reagent | APEX2_gBLOCK | Alice Ting lab, *Lam et al., 2015* | G block (double strand DNA fragment) | Sequence taken from Addgene pcDNA3 APEX2-NES (#49386) |
| sequence-based reagent | NASP_HDR_ upstream_F | This paper | PCR primers | AGCTACTCGCCCT GAACATGCA GAGCAGCACTG |
| sequence-based reagent | NASP_HDR_ upstream_R | This paper | PCR primers | CGTTCCCCTGAGG TGGCGAACCAGCGAACG |
| sequence-based reagent | GFPforNASP_HDR_F | This paper | PCR primers | TTCGCCACCTCAG GGGAACGATGGT GAGCAAGGGCGAGGA |
| sequence-based reagent | GFPforNASP_HDR_R | This paper | PCR primers | GCTGTGGACTCCAT GGCCATATGGCCC TGGAAGTAAAGGT |
| sequence-based reagent | NASP_HDR_ downstream_F | This paper | PCR primers | ATGGCCATGGAGT CCACAGCCACTGCCGC |
| sequence-based reagent | NASP_HDR_ downstream_R | This paper | PCR primers | TTATATTCCCGGGT TATCCAGGGGTTC TACCAGAGGCACACG |
| sequence-based reagent | NASP_pair1_upstream_F | This paper | DNA oligomer for dsDNA (guide RNA) | CACCGCCATGG CCATCGTTCCCCTG |
| sequence-based reagent | NASP_pair1_upstream_R | This paper | DNA oligomer for dsDNA (guide RNA) | AAACCAGGGGAA CGATGGCCATGGC |
| sequence-based reagent | NASP_pair1_ downstream_F | This paper | DNA oligomer for dsDNA (guide RNA) | CACCGAGCCAC TGCCGCCGTCGCCG |
| sequence-based reagent | NASP_pair1_ downstream_R | This paper | DNA oligomer for dsDNA (guide RNA) | AAACCGGCGACG GCGGCAGTGGCTC |
| sequence-based reagent | ASF1B_UP_R | This paper | PCR primers | GGCCATCGCC TCGCCTCGCC |
| sequence-based reagent | ASF1B_UP_F | This paper | PCR primers | AATCACTTCGG GTGCGAGCACC |
| sequence-based reagent | Spot-tag_ mCherry_3 C_R | This paper | PCR primers | GCACCGACACC TTGGCCATGGGCC CCTGGAACAGAAC TTCCAGGAGTCC GGACTTGTACAGCT |

*Continued on next page*

*Continued*

| Reagent type (species) or resource | Designation | Source or reference | Identifiers | Additional information |
|---|---|---|---|---|
| sequence-based reagent | Spot-tag_mCherry_3 C_F | This paper | PCR primers | GAGGCGAGGCGat ggccccggatcgcgtgcg cgcggtgagccattggag cagcGTGAGCAAGG GCGAGGAGGA |
| sequence-based reagent | ASF1B_DOWN_R | This paper | PCR primers | GGAAAATGGGAAG GGGCTGGATATTGG |
| sequence-based reagent | ASF1B_DOWN_F | This paper | PCR primers | ATGGCCAAGG TGTCGGTGC |
| sequence-based reagent | ASF1B_sgRNA_UP_F | This paper | DNA oligomer for dsDNA (guide RNA) | CACCGTCGCCTC GCCGCGCCGCAGC |
| sequence-based reagent | ASF1B_sgRNA_UP_R | This paper | DNA oligomer for dsDNA (guide RNA) | AAACGCTGCGGC GCGGCGAGGCGAC |
| sequence-based reagent | ASF1B_sgRNA_Down_F | This paper | DNA oligomer for dsDNA (guide RNA) | CACCGCAAGGTG TCGGTGCTGAACG |
| sequence-based reagent | ASF1B_sgRNA_Down_R | This paper | DNA oligomer for dsDNA (guide RNA) | AAACCGTTCAGC ACCGACACCTTGC |
| peptide, recombinant protein | Pierce High Capacity Streptavidin Agarose | Thermo Fisher | Cat. #: 20357 | |
| peptide, recombinant protein | GFP-trap agarose beads | ProteinTech (Chromoteck) | GTA | |
| chemical compound, drug | Biotin-phenol | Iris Biotech | LS-3500 | (Final concentration 500 µM) |
| chemical compound, drug | FugeneHD | Promega | E2311 | |
| chemical compound, drug | Rapamycin | ALFA (AESAR) | J62473.MC | (Final concentration 200 nM) |
| software, algorithm | R | R | Open Source | For graph plotting and statistical analysis |
| other | Hoechst 33342 stain | NEB | 4082 S | (1 µg/mL) |

## Cloning and plasmid material

Plasmids were constructed through cloning PCR, annealed oligo ligation, gBLOCK synthesis (IDT) and Gibson assembly (*Gibson, 2009*). All constructs were verified by Sanger sequencing. The open-reading frame from Imp5 isoform 1 (UniProt identifier: O00410-1) was amplified from HeLa cDNA and ligated into pEGFP-C1. For bacterial protein expression, Imp5 was re-cloned into pGEX-6P1 (Cytiva). (His)$_6$-sNASP, H3, and H4 histones constructs have been detailed previously (*Bowman et al., 2016*; *Bowman et al., 2017*). The RAPID-release plasmids were used as previously published (*Apta-Smith et al., 2018*). In order to contain the two-component RAPID-release system in a single vector, the tethered cassette (H3.2mCherry-TVMVcsx2-FRB-OMP25) and the protease cassette (FKBP12-TVMV-AI) were cloned either side of an internal ribosome entry site (IRES). CRISPR knock-in cell lines were obtained following the D10ACas9 nickase using the paired guide RNAs protocol (*Ran et al., 2013*). For the homologous DNA repair (HDR) template, 800 bp upstream and downstream of the start codon were amplified from genomic DNA, and ligated through Gibson assembly to EGFP and mCherry, respectively, so that an in-frame 5′ fusion to exon 1 was formed. This was cloned into pBlueScript II KS (+) as destination plasmid. Guide RNAs were prepared as described, combining two complementary oligonucleotides into D10ACas9 vector pX461-PSPCas9N(BB)–2A-GFP (Addgene #48140). For the inducible expression of H3-APEX-GFP, APEX2 was inserted between the H3 and GFP cassettes in the RAPID release vector pH3-eGFP$^{TVMVcsx2}$-FRB-OMP25 (*Apta-Smith et al., 2018*) and subcloned into pcDNAFRT/TO (Thermo) plasmid for recombination in the HeLa Flp-In T-REx system. See plasmid sequences in 'Key resources table' and primer sequences and DNA fragments in 'Key resources table' for details.

### Protein expression and purification

All recombinant proteins were produced in Rosetta2 *Escherichia coli* competent cells (Thermo Fisher) induced with 0.4 mM isopropyl-β-d-1-thiogalactoside (IPTG) at an $OD_{600}$ of 0.6, unless otherwise stated. sNASP was expressed overnight at 21°C as an N-terminal $(His)_6$ fusion construct and purified using Ni-NTA affinity resin (Cytiva) and further purified by ion-exchange chromatography after cleavage of the $(His)_6$ fusion by TEV protease as previously described (*Bowman et al., 2016*).

Imp5 purification was adapted from *Soniat et al., 2016*. Briefly, GST-Imp5 construct was induced for 12 hr at 24°C. Bacteria were pelleted and lysed in 50 mM Tris, pH 7.5, 150 mM sodium chloride, 20% glycerol, 2 mM dithiothreitol, 1 mM EDTA, protease inhibitors and lytic cocktail (10 µg/mL RNase A, 100 µg/mL DNase I, 1 mg/mL lysozyme) for 2 hr at 4°C, followed by flash-freezing in liquid nitrogen. Pellets were rapidly thawed at 37°C and quickly placed on ice before sonication with a Misonix sonicator (three pulses 'on' for 10 s and 'off' 30 s with an amplitude of 25). Imp5 was then purified with GST-trap resin (Cytiva) and further purified by anion-exchange chromatography following overnight cleavage with 3C protease.

Ran purification was performed as previously described, (*Padavannil et al., 2020*). Ran was GTP loaded at 10× final concentration for competition essays, with 50 mM GTP in 5 mM EDTA 50 mM HEPES pH 7.5 on ice for 30 min.

Full-length *Xenopus laevis* histones H3 and H4 were purified and refolded as previously described (*Luger et al., 1997*).

### Analytical gel filtration

Analytical gel filtration was carried out using a Superdex Increase 200 10/300 column (GE Healthcare) in 20 mM HEPES-KOH pH 7.5 and 150 mM sodium chloride, unless otherwise stated. 0.6 mL fractions were collected encompassing the void and bed volumes of the column. Proteins and complexes were reconstituted at a concentration of 20 µM. Fractions were separated by SDS-PAGE and stained with Coomassie Brilliant Blue (Instant Blue, Abcam).

### Competition assay

Purified Imp5 or sNASP were mixed with equimolar amount of histone H3 (20 µM final concentration) in 20 mM HEPES-KOH pH 7.5, 150 mM sodium chloride, 2 mM magnesium acetate, 1 mM EGTA, 2 mM dithiothreitol, and 10% glycerol and allow to equilibrate for 10 min on ice before spinning at 20,000 × *g* for 20 min at 4°C to remove any precipitate. Complexes were isolated on Superdex Increase S200 10/300 column (GE Healthcare) and central fractions were pooled, concentrated with a 10 kDa concentrator, before competition. Preformed Imp-H3 and sNASP-H3 complexes were combined with an equimolar amount of sNASP or Imp5, respectively, with or without the addition of Ran and GTP loaded Ran (RanGTP) and left to equilibrate for 3 hr on ice before separation on a glycerol gradient.

### Glycerol gradient ultracentrifugation

Adapted from *McClelland and McAinsh, 2009*. Briefly, protein mixtures were diluted to 6.25 µM in 200 µL of gradient loading buffer (20 mM HEPES pH 7.5, 150 mM sodium chloride, 2 mM dithiothreitol, 1% glycerol, and 5 mM GTP for the RanGTP samples) and gently layered on a 5 mL gradient of 5–40% glycerol (20 mM HEPES pH 7.5, 150 mM sodium chloride, 2 mM dithiothreitol 5 or 40% glycerol and 5 mM GTP for the RanGTP samples) made with Gradient Master (BioComp): Program Short Glycerol 5–40%. The samples were spun on a pre-chilled Optima XPN-80L ultracentrifuge (Beckman Coulter), rotor Sw55Ti, at 240,000 × *g* for 14 hr at 4°C. Then, 200 µL fractions were manually collected and separated by SDS-PAGE for Coomassie staining (Instant Blue, Abcam).

### Tissue culture and cell lines

HeLa Kyoto cells were sourced from CRUK Cell Services. HeLa FRT cell line (T-REx-HeLa) was sourced from Invitrogen (R71407). HEK 293F cells were sourced from Thermo Fisher (R79007). They were mycoplasma negative and were originally authenticated by short tandem repeat profiling. HeLa cells were grown to 70% confluency in DMEM-high glucose (Gibco) supplemented with 10% heat-inactivated foetal bovine serum (Sigma), 50 µg/mL penicillin/streptomycin at 37°C in a humidified incubator with 5% $CO_2$. Cells were passaged with 0.25% trypsin-EDTA and resuspended in complete

media. Cells were transfected following the FuGENE HD manufacturer's instructions (Promega) and imaged 24–48 hr post-transfection.

For stable lines, HeLa or HeLa FRT cells were transfected as described above. After 48 hr, antibiotic selection was performed and cells split when required. After 14 days, the transfection was sorted into fluorescent-positive single cells through flow cytometry.

For HEK293-F expression, cells were grown in suspension to $1 \times 10^6$ cells/mL with serum-free Free-Style 293 expression medium (Thermo Fisher), supplemented with 25 μg/mL penicillin/streptomycin, in a humidified incubator with 8% $CO_2$ and 120 rpm. HEK293-F cells were split 24 hr before transfection to $0.5 \times 10^6$ cells/mL, and transfected with 1 μg of DNA per $2 \times 10^6$ cells and 3 μg of linear PEI per 1 μg of DNA in OptiMEM (60 μL per 1 μg of DNA), vortexed briefly, and incubated at room temperature for 15 min before adding to the cells.

## CRISPR knock-in cell lines

CRISPR knock-in cells were obtained following the established protocol (*Ran et al., 2013*). Briefly, for HeLa cell lines, cells were co-transfected with plasmids encoding upstream and downstream sgRNA, the D10ACas9 nickase, and the HDR template on 6-well plates at 70% confluency using FuGENE HD (Promega). Cells were allowed to recover for 7–10 days (splitting if necessary) before single-clone selection through flow cytometry. For HEK293-F CRISPR knock-in populations, 10 mL of cells at a $1 \times 10^6$ cells/mL confluency and transfected with the same CRISPR plasmids as for HeLa cells following the transfection protocol for HEK293-F in suspension as described above. After 7 days (splitting when required), cells were fluorescence-activated cell sorted (FACS) to collect ↑ × $10^6$ cells, allow to recover for 15 days, and FACS again for purity ↑ × $10^6$ cells.

## Flow cytometry

Single HeLa, HeLa FRT, and HEK293-F cells were sorted with a BD FACSAria Fusion Flow Cytometer, using a 100 μm nozzle, gating FSC-A/FSC-H for singlets and laser compensation using negative and positive controls. HeLa and HeLa FRT cells were harvested by incubation with trypsin-EDTA for 5 min as before and rinsed in PBS twice by centrifugation at $200 \times g$ for 5 min at 4°C. Cells were then resuspended in phenol red-free DMEM supplemented with 2% FBS, filtered through a 50 μm Cell Strainer cap (Falcon) and sorted into 96-well plates with 1000 μL 1:1 fresh DMEM:conditioned DMEM (from pre-sorted medium) supplemented with 20% FBS. HEK293-F cells were resuspended in fresh FreeStyle 293 medium and sorted for yield. Sorted cells were rinsed in PBS twice by centrifugation at $300 \times g$ for 5 min at 4°C. Final media contained 1:1 fresh:conditioned (from pre-sorted) medium.

## Cell fractionation

Cell fractionation was performed following the REAP protocol (*Suzuki et al., 2010*) as previously adapted by *Apta-Smith et al., 2018*. Either cells on plates or cells in suspension were washed twice with PBS before collection. Cells on suspension were pelleted at $200 \times g$, 5 min, 4°C between washes. Adherent cells were scraped into ice-cold PBS and pelleted the same way before resuspension in lysis buffer (0.1% NP-40 in PBS containing the protease inhibitors PMSF, aprotinin, leupeptin, pepstatin, and benzamidine). Lysis was continued for 10 min on ice, before pelleting of the chromatin fraction for 10 min at $1000 \times g$, 4°C. The supernatant was again spun at $2000 \times g$, 4°C, and then removed to a fresh tube and spun for 20 min at $20,000 \times g$ to remove cell debris. For Co-IP experiments, in order to ensure identical loading across samples, the amount of GFP-bound protein was assessed by loading final extract in sample buffer (without boiling) on a Mini-PROTEAN TGX Precast Gel (Bio-Rad), visualised with LED$_{488}$ illumination (G-box, Syngene) and quantified against standard curve of recombinant GST-eGFP using ImageJ software (*Figure 1—figure supplement 1*).

## Immunoprecipitation

Soluble cell fractions were bound to 10 μL per mL of lysate of GFP-Trap (ChromoTek) agarose beads. Binding was allowed to proceed for 1 hr at 4°C with continual nutation. Beads were spun for 2 min at $250 \times g$, 4°C, and washed once in 20 mM Tris–HCl pH 7.5, 400 mM sodium chloride, and 0.1% NP-40 (high salt buffer), followed by three washes in PBS + 0.1% NP-40, and one last wash in PBS. For SDS-PAGE analysis, the beads were either boiled directly in 2× Laemmli sample buffer. For 1D Native PAGE, the beads were incubated in PBS 2 μM dithiothreitol at 4°C for 16 hr with TEV protease to

release bait complexes. For mass spectrometry, the beads were washed four times with 50 mM ammonium bicarbonate, alkylated in 10 mM TCEP, 40 mM CAA in 50 mM ABC for 5 min at 70°C, and incubated with 1 µg trypsin per 100 µg protein and digested at 37°C overnight.

## Glycerol gradient ultracentrifugation

After overnight TEV cleavage, soluble samples were separated from GFP-trap matrix by filtration through 0.22 µm nylon filter (Costar) spun at 2000 × *g* for 2 min at 4°C before directly loading on a glycerol gradient for separation (*McClelland and McAinsh, 2009*). As described above, IP extracts were gently layered on a 5 mL gradient of 4–20% glycerol (20 mM HEPES pH 7.5, 150 mM sodium chloride, 2 mM dithiothreitol) made with Gradient Master (BioComp): Program Short Glycerol 5–25% w/v. The samples were spun on a pre-chilled Optima XPN-80L ultracentrifuge (Beckman Coulter), rotor Sw55Ti, at 240,000 × *g* for 14 hr at 4°C. Then, 200 µL fractions were manually collected and separated by SDS-PAGE for Coomassie staining (Instant Blue, Abcam) and Western blot analysis.

## Native gel (1D) and 2D gel separation

After overnight TEV cleavage, soluble samples were separated from GFP-trap matrix by filtration through 0.22 µm nylon filter (Costar) spun at 2000 × *g* for 2 min at 4°C. Protein complexes were concentrated to 25 µL in 30 kDa centrifugal concentrator (Vivaspin 500 Sartorius) and mixed with equal volume of 2× native gel sample buffer (62.5 mM Tris–HCl, pH 6.8, 40% glycerol, 0.01% bromophenol blue) (Bio-Rad). Three lanes were loaded on a 4–20% polyacrylamide Mini-PROTEAN TGX Precast Gel (Bio-Rad) and run with 25 mM Tris, 192 mM glycine, pH 8.4 buffer. The gel was run for 6 hr below 10 mA (20 min at 20 V, 30 min at 30 V, 60 min at 60 V, 120 min at 120 V, and 120 min at 160 V). The gel lanes were longitudinally cropped, using one of them for Coomassie staining, from which bands were cut and stored at –80°C until mass spectrometry analysis. Gel bands were in-gel digested according to *Shevchenko et al., 2006*. The other two lanes were incubated with 1× Tris/glycine/SDS buffer for 15 min and then incubated with 2× Laemmli sample buffer (Bio-Rad) for 15 min, heated twice for 15 s in the microwave (800 mA), and loaded on 15% polyacrylamide gel for 2D separation. One gel was dedicated to Coomassie staining and the other one was used for Western blot analysis.

## Western blot analysis

SDS-PAGE gels were transferred onto a nitrocellulose membrane using an iBlot 2 Dry Blotting System (Thermo Fisher), blocked with 3% powdered milk in TBS-T, and then incubated at 4°C overnight with primary antibodies diluted in 3% powdered milk or 3% BSA. Antibody dilutions were made at 1:1000 ratios except anti-NASP (*Apta-Smith et al., 2018*, which was diluted to 1:10,000–40,000). Membranes were washed with TBS-T and then incubated for 1 hr at room temperature with the HRP-conjugated secondary antibody. See 'Key resources table' for a list of antibodies.

## APEX biotinylation and sample preparation

Proximity biotinylation was performed according to *Lam et al., 2015*. Briefly, per condition, four 15 cm plates were induced for 24 hr with 0.5 µg/mL doxycycline. Rapamycin (200 nM final) was added 120 min before harvesting to release histone H3-APEX2-EGFP compared to mock-treated cells (time 0). Thirty minutes before harvesting, cell media was supplemented with biotin-phenol (Iris Biotech) to final concentration of 500 µM and incubated for 30 min at 37°C 5% $CO_2$, before 200 µL per 15 cm plate of 100 mM $H_2O_2$ (final concentration 1 mM) was added and incubated for 1 min at room temperature with gentle shaking (100 mM stock was prepared from fresh $H_2O_2$ Merck 30% w/w).

The reaction was quenched by washing cells three times with ice-cold 5 mM trolox, 10 mM sodium ascorbate, 0.03% sodium azide in PBS. Cells were scraped off the plates and lysed with RIPA lysis buffer (50 mM Tris–HCl, pH 7.5, 150 mM sodium chloride, 0.1% SDS, 0.5% sodium deoxycholate, 1% Triton X-100, protease inhibitors [as detailed above] and 2 µL benzonase [Sigma]) for 20 min on ice. Lysates were clarified by centrifugation at 16,000 × *g* for 10 min at 4°C before protein concentration was determined by Bradford assay. Then, 25 µL of Pierce High Capacity Streptavidin Agarose (Thermo Fisher) was used for each condition. Lysates of equal protein concentrations were incubated with streptavidin beads for 60 min at 4°C, washed twice with cold lysis buffer, once with cold 1 M KCl, once with cold 100 mM sodium carbonate, once with cold 2 M urea in 50 mM ammonium bicarbonate, and three times in 50 mM ammonium bicarbonate.

Samples were alkylated in 10 mM TCEP, 40 mM chloroacetamide in 50 mM ammonium for 5 min at 70°C and digested with 1 μg trypsin per 100 μg protein at 37°C overnight. The pH was adjusted to 3.5 before peptide storage at –20°C.

## LC-MS/MS and data analysis

Peptides from on-gel digestion or on-beads digestion were analysed on a Orbitrap Fusion with Ulti-Mate 3000 RSLCnano System (Thermo Scientific). The raw data were searched using MaxQuant software (version 1.6.10.43) against UniProtKB Human database and the common contaminants database from MaxQuant (*Tyanova et al., 2016*) from peptides generated from a tryptic digestion.

The software Scaffold (version 4.8.9, Proteome Software Inc) was used to validate MS/MS-based peptide and protein identifications. With the exception of BioID experiments, peptide identifications were accepted if they could be established at 95% probability or higher by the Scaffold Local FDR algorithm. Protein identifications were accepted if they could be established at 99% probability or higher and contained at least two identified peptides. For BioID experiments, thresholds were 95% for peptide identification, 95% for protein identification. Proteins that contained similar peptides and could not be differentiated based on MS/MS analysis alone were grouped to satisfy the principles of parsimony. Proteins sharing significant peptide evidence were grouped into clusters. For the histone mutant experiments, since the data was very robust with strong signal, normalised total precursor intensity for proteins or protein clusters containing that were not identified in all three replicates of at least one condition (WT and the two mutants) were filtered out. Next 0 values were replaced with a fixed value below detection limit, arbitrarily assigned to 1/2 of the lowest detected value. The $\log_2$ of these values were normalised with 'scale' Limma package normalisation and used to build a linear model (*Ritchie et al., 2016*) from which adjusted p-values (FDR) were obtained for pairwise comparisons (Table S4). For the native gel protein identification, ANOVA was performed in R with adjusted p-values (FDR) (Table S4). For the BioID experiments, containing more protein hits and far more 0 values than the affinity-purified experiments, they were processed with DEP package (Zhang 2018, 10.1038/nprot.2017.147) for data normalisation and imputation with the parameters ("MinProb" q=0.01), recommended for samples with numerous missing values (Table S4), from which a linear model with Limma used DEP directly.

The mass spectrometry proteomics data have been deposited to the ProteomeXchange Consortium via the PRIDE (*Perez-Riverol et al., 2019*) partner repository with the dataset identifier PXD029354 and 10.6019/PXD029354.

## Imaging

All images were captured using an UltraVIEW VoX Live Cell Imaging System (PerkinElmer) with a 37°C environmental chamber. Cells were cultured in 8-well μ-slides (ibidi) with 200 μL Leibovitz's L-15 (L-15) medium (Thermo Fisher Scientific) supplemented with 4 mM L-glutamine and 50 μg/mL penicillin/streptomycin. For DNA staining, cells were stained with 1 μg/mL Hoechst for 10 min before washing with PBS and restoration of L-15 medium.

For RAPID-release experiments, L15 medium containing 1 μM rapamycin was added directly to the well, resulting in a final concentration of 200 nM rapamycin. An initial image stack was taken prior to rapamycin addition, serving as time 0, after which cells were imaged every minute for up to 30 min.

For RAPID-release of tethered H3.1-mCherry and soluble eGFP-Imp5, cells were co-transfected with H3.1-mCherry-RR and EGFP-Imp5 with release being performed as described above, allowing H3.1 and Imp5 to be followed in the red and green channels, respectively. For RAPID-release of tethered H3.1-mCherry and endogenous EGFP-NASP, a stable cell line was used.

Partitioning and quantification of the images were carried out using ImageJ. Z-stacks spanning the cell were flattened into a maximum pixel intensity image. The cytoplasm and nucleus were manually partitioned for each cell, and the nuclear enrichment over the cytoplasm was calculated for each time point. Values for individual cells were normalised between 1 and 0 and plotted on the same axes for comparison.

The Pearson's correlation coefficients for Mito-2-hybrid (M2H) between the mCherry and the eGFP channels were calculated at each time point using the ImageJ GDSC Colocalisation Threshold plugin.

## FRAP

H3.1-/H4-EGFP wild-type and mutants (FD, HB) were transfected into HEK293-F cells and seeded onto 8-well μ-slides (ibidi). FRAP experiments were performed using an UltraVIEW VoX Live Cell Imaging System (PerkinElmer) equipped with a PhotoKinesis FRAP module. Four pre-bleaching images of a single z-slice were taken before bleaching circular regions of 2 μm diameter in the cell nucleus using 30 iterations of the 405 nm laser at 50% power. Images were taken directly after bleaching and every 15 s up to 4 min. Image analysis was performed in ImageJ. Values were normalised, taking the average of pre-bleach images as 1 and the lowest post-bleach image as 0. Error bars represent the SD of at least four individual experiments, containing each 2–5 bleached cells.

## Acknowledgements

Facility support was provided by the Warwick Proteomics Research Technology Platform, CAMDU (Computing and Advanced Microscopy Development Unit), and Warwick Integrative Synthetic Biology (WISB) Centre (for access to flow cytometry facilities). We would like to acknowledge Andrew Bottrill, Cleidi Zampronio, and Claire Mitchell for technical support, and colleagues in the Division of Biomedical Sciences at Warwick Medical School for support and advice. This work was supported by a Wellcome Trust Sir Henry Dale Fellowship to AJB (208801/Z/17/Z).

## Additional information

### Funding

| Funder | Grant reference number | Author |
| --- | --- | --- |
| Wellcome Trust | 208801/Z/17/Z | Alonso Javier Pardal<br>Andrew James Bowman |

The funders had no role in study design, data collection and interpretation, or the decision to submit the work for publication. For the purpose of Open Access, the authors have applied a CC BY public copyright license to any Author Accepted Manuscript version arising from this submission.

### Author contributions

Alonso Javier Pardal, Conceptualization, Validation, Investigation, Visualization, Methodology, Writing - original draft; Andrew James Bowman, Conceptualization, Funding acquisition, Investigation, Methodology, Writing - original draft, Writing - review and editing

### Author ORCIDs

Andrew James Bowman http://orcid.org/0000-0002-5507-669X

### Decision letter and Author response

Decision letter https://doi.org/10.7554/eLife.81755.sa1
Author response https://doi.org/10.7554/eLife.81755.sa2

## Additional files

### Supplementary files

- Transparent reporting form
- Source data 1. Uncropped source images for gels and western blots.

### Data availability

Raw proteomics data have been deposited to the ProteomeXchange Consortium via the PRIDE partner repository with the dataset identifier PXD029354 and https://doi.org/10.6019/PXD029354. Analysed proteomic results are provided as source data files, as referred to in the main text. Uncropped, source images for western blots and gels are provided as a Supplementary Material.

The following dataset was generated:

| Author(s) | Year | Dataset title | Dataset URL | Database and Identifier |
|---|---|---|---|---|
| Bowman AJ, Pardal A | 2021 | Human cell lines pulldown interactions CL-MSMS | https://www.ebi.ac.uk/pride/ | PRIDE, PXD029354 |

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
