## [Editor Report]

The authors provide good evidence showing that a pool of newly synthesised H3 and H4 histones are maintained in a monomeric conformation in the cytoplasm and are translocated into the nucleus via the importin-5 protein. The analysis makes great use of various technologies, and the results are certainly interesting.

---

## [Decision Letter]

**Decision letter after peer review:**

Thank you for submitting your article "A specific role for Importin-5 and NASP in the import and nuclear hand-off of monomeric H3" for consideration by *eLife*. Your article has been reviewed by 3 peer reviewers, including Jerry L Workman as Reviewing Editor and Reviewer #1, and the evaluation has been overseen by Kevin Struhl as the Senior Editor. The following individuals involved in the review of your submission have agreed to reveal their identity: Hongda Huang (Reviewer #2); Eric Campos (Reviewer #3).

Essential revisions:

The article is interesting because it touches on the underappreciated handling of new histones as monomers, and explains how the monomeric histones are imported into the nucleus. It is also interesting from a molecular point of view because histone monomers may perhaps add an additional regulatory mechanism by which different H3 variants are assembled with H4 at the right time (perhaps something for the authors to look into in future studies).

The goal of showing how monomeric histones are handled is achieved, but their interpretation of previously published results is too often dismissed through hand waving (e.g., subcellular fractionations are inherently flawed), or speculative (untested) statements (e.g., the H3-H4-ASF1-Imp4 complex previously identified by various groups, represents nuclear histones that diffused out of the nucleus). There are important prior findings that would also be interesting to discuss (some examples are listed below). That said, it should be emphasized that the authors' findings are novel and very interesting.

Experimental Recommendations:

1. Figure 3: The level of protein overexpression should be indicated. If Figure S2E shows a western from cell extracts (it presumably is, but the panel is not included in the figure legend), there is then a high degree of overexpression, which should be discussed.

2. Figure 3D: There are multiple bands on the silver-stained gel with a molecular weight near that of the band labeled as sNASP, but they don't all have the same migration pattern on the glycerol gradient. One of these bands should be UBR7 but were any of the other bands NASP as well (if not, what were they)?

3. It may be worth showing more of the matching western blot (cropped very close to the sNASP signal) to show if any other NASP bands appeared below (or not).

Recommendations regarding interpretation and points of discussion:

4. The cytoplasmic pool of histones is very low, which is why prior biochemical purifications from the cytoplasm started with large amounts of extracts. Yes, subcellular fractionations suffer from some cross-contamination, but that does not explain why histones isolated from cytoplasmic sNASP largely lacked PTMs seen in nucleosomal histones (MS analysis in the cited NSMB 2010).

5. Speculation on ASF1-Imp4 solely associating with evicted nucleosomal histones that diffuse to the cytoplasm is not particularly convincing. That would require formal testing, and it could very well be that there is more than one way to import histones into the nucleus.

6. Figure 3E-G: It's curious that the majority of sNASP was not associated with any histones. Could this be due to the overexpression?

7. How do HSPA8 and other prior data fit into the monomeric histone model? Do you, for example, foresee monomeric histones being modified on ribosomes as demonstrated by the Loyola lab, folded by HSP8, and directly transferred to Imp5?

8.- The pitfalls of protein overexpression/co-overexpression, following mutant histones, and of artificially tethering large amounts of histones to the mitochondrial outer membrane could be better discussed (these are really cool techniques, but they of course have their limitations).

9. Similarly, overexpressed, mitochondrial-tethered APEX2-tagged protein should not be said to represent 'endogenous conditions'. (Though it certainly is a great use of technology!)

10. Discuss some of the limitations, especially regarding the existence of a soluble pool of monomeric H4 and their functional relevance in vivo.

11. RBBP7 and RBBP4 are also called RBAP46 and RBAP48, respectively. Please make this clear to help readers.

---

## [Author Response]

Essential revisions:The article is interesting because it touches on the underappreciated handling of new histones as monomers, and explains how the monomeric histones are imported into the nucleus. It is also interesting from a molecular point of view because histone monomers may perhaps add an additional regulatory mechanism by which different H3 variants are assembled with H4 at the right time (perhaps something for the authors to look into in future studies).The goal of showing how monomeric histones are handled is achieved, but their interpretation of previously published results is too often dismissed through hand waving (e.g., subcellular fractionations are inherently flawed), or speculative (untested) statements (e.g., the H3-H4-ASF1-Imp4 complex previously identified by various groups, represents nuclear histones that diffused out of the nucleus). There are important prior findings that would also be interesting to discuss (some examples are listed below). That said, it should be emphasized that the authors' findings are novel and very interesting.

We thank the reviewers for their time and constructive feedback. We have addressed the issues raised and have submitted a revised version detailing our changes, which we hope will satisfy the reviewers’ concerns.

Experimental Recommendations:1. Figure 3: The level of protein overexpression should be indicated. If Figure S2E shows a western from cell extracts (it presumably is, but the panel is not included in the figure legend), there is then a high degree of overexpression, which should be discussed.

We thank the reviewer for bringing this to our attention. Figure S2E is indeed a western from whole cell extracts, but it is from the CRISPR eGFP knock in HeLa cell line shown in panels A, B, C, and D of the same figure. The legend was erroneously truncated in the version submitted, a full legend explaining the figure is now included in the revised version Figure 3 - figure supplement 2. In addition, we have now included a new supplementary figure (new Figure 3 -figure supplement 1 in the revised manuscript) showing the level of protein expression in transiently transfected HEK293-F cells.

2. Figure 3D: There are multiple bands on the silver-stained gel with a molecular weight near that of the band labeled as sNASP, but they don't all have the same migration pattern on the glycerol gradient. One of these bands should be UBR7 but were any of the other bands NASP as well (if not, what were they)?

The reviewer points to several bands around the predominant sNASP species that differ in elution profiles. These most likely represent the co-chaperoning factors identified in the mass spec analysis (HAT1 (49.5 kDa), RBBP7 (47.8 kDa), UBR7 (48 kDa), and HSP70 proteins). As they all migrate at similar positions in SDS-PAGE, we performed western blots to identify a number of them individually. These complexes migrate with H3-H4, and are thus likely co-chaperoning complexes of NASP which have been identified previously by others, and which we separate at a higher resolution in Figure 3E and G. We think we may have caused confusion by omitting a label indicating which panels were western blots within Figure 3D (although it was noted in the legend). To address this, we have added a label to the lefthand side of the figure in the revised version.

3. It may be worth showing more of the matching western blot (cropped very close to the sNASP signal) to show if any other NASP bands appeared below (or not).

The full uncropped western blot for this figure (along with other westerns) is shown in the supporting material file. We do see some minor lower molecular weight bands on a long exposure. However, these most likely represent low level proteolytic degradation due to NASP having significant disordered regions. We have now shown a larger portion of the cropped area in the main Figure 3D.

Recommendations regarding interpretation and points of discussion:4. The cytoplasmic pool of histones is very low, which is why prior biochemical purifications from the cytoplasm started with large amounts of extracts. Yes, subcellular fractionations suffer from some cross-contamination, but that does not explain why histones isolated from cytoplasmic sNASP largely lacked PTMs seen in nucleosomal histones (MS analysis in the cited NSMB 2010).

We agree with the reviewer’s referenced work that sNASP-bound histones do not contain nucleosomal modifications. In our eyes the question is not whether these cytoplasmic or nucleosomal, but whether they are cytoplasmic or nucleoplasmic. Immunostaining and GFP fusions show that NASP is nuclear, whereas cell fractionation shows it is cytosolic. We attempted to reconcile this in Apta-smith et al., 2018. Our results here agree with the work of Campos et al., 2010, in that NASP is an early binder of newly synthesised H3, interacting before ASF1 and HAT1-RBBP7, but we propose that this happens in the nucleus as NASP is an overwhelmingly nuclear protein.

5. Speculation on ASF1-Imp4 solely associating with evicted nucleosomal histones that diffuse to the cytoplasm is not particularly convincing. That would require formal testing, and it could very well be that there is more than one way to import histones into the nucleus.

We agree with the reviewer that we have emphasised this too much in the discussion and model, and, although it is a possibility, we don’t provide any evidence for it. We have removed this from the model (Figure 7) and represented it more clearly in the text in the revised discussion, and curtailed our speculation whilst still indicating it as a possible explanation:

Discussion, page 10:

“If newly synthesised H3 and H4 are rapidly imported into the nucleus as monomers bound to Imp5, what is the role of the long-established Imp4-H3-H4-ASF1 complex? The most likely explanation would be a parallel pathway for importing H3 and H4 that have folded in the cytoplasm {campos_new_2010, alvarez_sequential_2011}. An alternative scenario could be that Imp4 recognises post-nuclear H3-H4-ASF1, promoting the re-import of H3-H4 dimers that may have escaped. Indeed, the H3-H4-ASF1 complex would be small enough to transit the nuclear pore passively {frey_surface_2018}, with ASF1, unlike NASP, not containing a nuclear localisation sequence.”

6. Figure 3E-G: It's curious that the majority of sNASP was not associated with any histones. Could this be due to the overexpression?

It is an interesting observation. We don’t think it is due to overexpression per se, as the CRISPR eGFP knock-in (which should represent close to endogenous levels of NASP driven by its endogenous promoter) had a significant proportion of free sNASP compared to histone-bound sNASP (Figure 3G). The reviewer is correct, however, in that this may be exacerbated somewhat in the overexpression line used for proteomic analysis (Figure 3E). To address this, we have performed an anti-NASP western blot on the HEK293-F cells overexpressing eGFP-sNASP and find that the level of expression is modest compared the endogenous level. This is represented as a new Figure 3—figure supplement 1.

As the complex profiles are very similar between Figure 3E (overexpression) and Figure 3G (endogenous) we don’t think the over-expression is affecting complex formation, whose characterisation was the purpose of this figure, but may have effects on the bulk levels of soluble histone as seen in Cook et al., 2011. In our eyes it would make sense that chaperoning proteins are in excess of their histone substrate so that there is significant room to buffer fluctuations in free histone levels, especially upon, for example, acute replication stress as has been shown by others.

7. How do HSPA8 and other prior data fit into the monomeric histone model? Do you, for example, foresee monomeric histones being modified on ribosomes as demonstrated by the Loyola lab, folded by HSP8, and directly transferred to Imp5?

The role of HSPA8 and other HSP70 family chaperones in histone folding pathway is interesting, but in our eyes these interactions are difficult to interpret as they occur in numerous complexes and may rather relate to the likelihood of hydrophobic regions of histones being exposed in any one complex rather than a predetermined pathway. Campos et al., 2010 and Alvarez et al., 2011 isolated HSP70 proteins in complex with monomeric H3, with our APEX2 data showing HSPA8 (HSP70) associates with monomeric H3 tethered in the cytoplasm, reinforcing these findings. However, we also see it binding to NASP-H3-H4 complexes alongside other HSP70 members (HSPA1A and HSPA5). Further, recent findings from the Groth lab have identified DNAJC9 as a specific histone binding protein which aids in the recruitment of HSP70 proteins to histones (Hammond et al., 2021). We have now discussed these points in the paragraph starting -

“With regards to integration with the heat-shock protein folding machinery, The presence of HSPA8 bound to cytoplasmically tethered H3 is in agreement with previous findings that show HSP70 proteins bind to newly synthesised H3 before folding with H4 {campos_new_2010, alvarez_sequential_2011}.”

In terms of the Loyola lab’s finding that H3 K9 is methylated on ribosomes by SETDB1, we don’t see any reason why this would not also be the case for H3 imported as a monomer via the Imp5 pathway. But we also don’t see any reason why it would be necessary for import as it is predominantly a pre-deposition mark for heterochromatin, and therefore didn’t explicitly expand on it in the discussion.

8.- The pitfalls of protein overexpression/co-overexpression, following mutant histones, and of artificially tethering large amounts of histones to the mitochondrial outer membrane could be better discussed (these are really cool techniques, but they of course have their limitations).

We accept that the manuscript may not have proportionally discussed the limitations of the techniques employed now included a new subsection “Limitations of the Study” on these limitations at the end of the “Discussion” section.

9. Similarly, overexpressed, mitochondrial-tethered APEX2-tagged protein should not be said to represent 'endogenous conditions'. (Though it certainly is a great use of technology!)

We agree with the reviewers and have modified the text to reflect this. We were attempting to portray that the prey proteins in the APEX2 experiment were at endogenous ‘levels’, rather than overexpressed in the mito-2-hybrid experiment, but realise our wording was not clear in this instance.

10. Discuss some of the limitations, especially regarding the existence of a soluble pool of monomeric H4 and their functional relevance in vivo.

This has been included in the “Limitations of the Study” subsection at the end of the “Discussion” section.

11. RBBP7 and RBBP4 are also called RBAP46 and RBAP48, respectively. Please make this clear to help readers.

We appreciate the reviewers bringing our attention to this point. We have highlighted this in their first appearance in the manuscript.